# ACCELERATING DATA GENERATION FOR NONLINEAR TEMPORAL PDEs VIA HOMOLOGOUS PERTURBATION IN SOLUTION SPACE

## ABSTRACT

Data-driven deep learning methods like neural operators have advanced in solving nonlinear temporal partial differential equations (PDEs). However, these methods require large quantities of solution pairs—the solution functions and right-hand sides (RHS) of the equations. These pairs are typically generated via traditional numerical methods, which need thousands of time steps iterations far more than the dozens required for training, creating heavy computational and temporal overheads. To address these challenges, we propose a novel data generation algorithm, called HOmologous Perturbation in Solution Space (HOPSS), which directly generates training datasets with fewer time steps rather than following the traditional approach of generating large time steps datasets. This algorithm simultaneously accelerates dataset generation and preserves the approximate precision required for model training. Specifically, we first obtain a set of base solution functions from a reliable solver, usually with thousands of time steps, and then align them in time steps with training datasets by downsampling. Subsequently, we propose a "homologous perturbation" approach: by combining two solution functions (one as the primary function, the other as a homologous perturbation term scaled by a small scalar) with random noise, we efficiently generate comparable-precision PDE data points. Finally, using these data points, we compute the variation in the original equation's RHS to form new solution pairs. Theoretical and experimental results show HOPSS lowers time complexity. For example, on the Navier-Stokes equation, it generates 10,000 samples in approximately 10% of traditional methods' time, with comparable model training performance.

## 1 INTRODUCTION

Nonlinear temporal partial differential equations (temporal PDEs) serve as a core mathematical tool for precisely characterizing continuous physical systems in the real world that evolve dynamically over time. They possess exceptionally high application value across diverse real-world scenarios: for instance, the Navier-Stokes equations (Temam (2024)) describe the motion states of atmospheric fluids. Traditionally, solving PDEs has often relied on extensive domain expertise and computationally intensive numerical methods (e.g., the finite difference method Godunov & Bohachevsky (1959), finite element methods Strang et al. (1973)). With the rapid advancement of deep learning, a new physical dynamics modeling and prediction paradigm, such as neural operators, has sparked widespread discussion. Deep learning models can unearth latent physical relationships from data and predict future states at lower computational cost—a capability driving numerous breakthroughs in physical dynamics research, modeling, and prediction.

Neural operators—especially transformer-based ones—show promise for accelerating PDE solutions. But they rely heavily on large-scale training data, which comes from costly classical methods (e.g., FEM Hughes (2000)). This creates a hard-to-solve circular dependency issue (Brandstetter et al. (2022)). This dependency becomes particularly problematic at industrial scales, as the high computational overhead of data generation via traditional numerical solvers severely hinders real-world applications (Zhang et al. (2025)). By accelerating dataset generation, we not only mitigate the high costs of classical method-driven model order reduction but also enhance sample complexity. Furthermore, efficient data generation supports broader generalization capabilities, which remains

an underexplored area critical for robust real-world applications (Kochkov et al. (2021), Stachenfeld et al. (2021)).

However, existing methods for generating PDE datasets exhibit notable limitations. Traditional numerical methods (e.g., the Crank-Nicolson method) typically require iterating over thousands of time steps to achieve solution convergence and stability, incurring significant temporal and computational overheads. While progress has been made in addressing linear time-independent PDEs (e.g., DiffOAS Dong et al. (2024), Wang et al. (2024)), these approaches rely on the linearity and time-invariance of the equations. This renders them incompatible with the data needs of neural solvers for nonlinear temporal PDE scenarios.

To address the challenges in forward modeling for nonlinear temporal PDEs, we propose a novel and efficient data generation algorithm termed HOmologous Perturbation in Solution Space (HOPSS). HOPSS simultaneously accelerates dataset generation while preserving the precision of the generated data. Specifically, HOPSS first generates a set of solution pairs under initial conditions, which serves as base solution functions in the solution space. These base solution functions are typically generated using a traditional solver to ensure high precision, and are then aligned to the time steps required for model training. Next, the base functions are fed into the generator, where two solutions are randomly selected: One is scaled by a small scalar, acting as the homologous perturbation term. It is then combined with the other to generate a new solution, plus small random noise. Concurrently, HOPSS computes the right-hand side (RHS) of the physical equation, ensuring compliance with the properties of the corresponding physical equation. A key advantage of HOPSS lies in its ability to avoid extensive iteration over a large number of time steps; instead, it acts directly on the time steps needed for training.

The distinct contributions of our work can be summarized as follows.

- We propose a novel data generation algorithm (HOPSS) tailored for nonlinear temporal PDEs. This algorithm simultaneously accelerates dataset generation while preserving the precision of the generated data—enabling the generation of large-scale nonlinear temporal PDE datasets at affordable time costs and computing resources.

- We demonstrate that our proposed algorithm significantly lowers computational complexity and shortens data generation time when solving nonlinear temporal PDEs. Notably, even with only approximately 10% of the generation time required by existing methods, neural operators trained on HOPSS-generated data exhibit performance comparable to those trained on data from conventional generation approaches.

## 2 RELATED WORK

### 2.1 DATA-DRIVEN DEEP LEARNING FOR SOLVING TEMPORAL PDES

Data-driven deep learning has become a transformative force in solving temporal PDEs, with key advancements centered on efficient model architectures and hybrid computational paradigms. Neural operators stand out as a major breakthrough: the Fourier Neural Operator (FNO Li et al. (2021)) and Deep Operator Network (DeepONet Lu et al. (2021))leverage deep learning to capture complex spatiotemporal dependencies in PDE systems, outperforming conventional methods in efficiency for time-evolving problems. Complementary efforts target bottlenecks in classical PDE solving—for instance, studies exploring neural networks to accelerate linear equation system solutions directly reduce the computational overhead of temporal PDE workflows. Additionally, data-driven solvers have advanced via hybrid designs: works like Hsieh et al. (2019), Yang et al. (2016), and Kochkov et al. (2021) propose data-optimized iterative schemes, merging machine learning with traditional numerical techniques to boost efficiency for temporal PDE scenarios. Together, these directions highlight a shift toward data-centric methods that address the unique spatiotemporal challenges of temporal PDEs.

### 2.2 DATA GENERATION FOR PDE ALGORITHMS

Training data-driven PDE algorithms, particularly those targeting nonlinear temporal PDEs, demands large-scale offline paired parametrized datasets. These datasets must capture complex spatiotemporal dynamics—including nonlinear interactions (e.g., convection terms in Navier-Stokes

equations) and transient behaviors—making their generation computationally intensive. Traditionally, such datasets are produced exclusively via classical computational mathematics algorithms, following a standardized numerical workflow tailored to solve nonlinear temporal PDEs.

Numerically solving nonlinear temporal PDEs typically follows the method of lines approach. This involves two principal steps: First, spatial discretization is applied to approximate the spatial derivatives. Core techniques such as the Finite Difference , Finite Element , Finite Volume Methods Strikwerda (2004); Hughes (2000); LeVeque (2002) are employed, each suited for different domain geometries and conservation properties. This process transforms the continuous PDE into a system of time-dependent ordinary differential equations (ODEs) defined on the spatial grid. Second, temporal discretization is used to integrate the resulting ODE system over time. For this, a suitable time-stepping scheme is chosen. Explicit schemes (e.g., the 4th-order Runge-Kutta method Hairer et al. (1993)) are often used for non-stiff problems due to their simplicity, while implicit schemes (e.g., the Crank-Nicolson method Crank & Nicolson) are preferred for stiff systems to overcome stringent stability constraints on the time step. A common challenge with implicit schemes for nonlinear equations is the treatment of nonlinear terms, which often necessitates linearization procedures (e.g., Newton-Raphson iteration Crisfield (1979)) at each time step to form solvable linear systems. It is important to note that alternative strategies, such as Implicit-Explicit (IMEX Burman et al. (2024)) schemes, can handle certain nonlinearities without full linearization by splitting the equation into implicitly and explicitly treated parts. While traditional numerical methods for generating nonlinear temporal PDE datasets are highly mature, they demand substantial time and computational resources—largely due to the repeated solution of multiple matrix multiplications at each time step (Hao et al. (2022)). This inefficiency poses a significant bottleneck to the advancement of data-driven deep learning algorithms for PDEs, as large-scale dataset generation often becomes prohibitively costly. To address this challenge, emerging data generation methods have been proposed, with DiffOAS (Dong et al. (2024)), Wang et al. (2024), and Brandstetter et al. (2022) being representative examples. However, DiffOAS is specifically designed for data generation of linear time-independent PDEs; it cannot be directly extended to nonlinear temporal PDEs. A critical limitation is that DiffOAS relies heavily on the linear properties and steady-state characteristics of the target equations. This dependence renders it inadequate for nonlinear temporal scenarios, as it fails to preserve the spatiotemporal dynamics and physical precision required for training reliable data-driven solvers (Dong et al. (2024)).

## 3 PRELIMINARIES

### 3.1 NONLINEAR TEMPORAL DATA GENERATION

Our primary goal is to generate nonlinear temporal datasets, which are acquired by solving associated partial differential equation (PDE) problems. Numerical methods for PDE solving are categorized into explicit and implicit schemes; these methods discretize PDE problems by mapping them from infinite-dimensional function spaces to finite-dimensional spaces, ultimately resulting in either a system of linear equations or a sequence of matrices. To illustrate this, we take the Navier-Stokes equations 1 and the Crank-Nicolson method—a classic implicit scheme—as an example.

$$u_t + \mathbf{v} \cdot \nabla u = \nu \nabla^2 u + f, \tag{1}$$

Generating datasets for temporal PDEs typically relies on the Crank-Nicolson (CN) scheme—an implicit temporal discretization method ideal for balancing stability and efficiency. Its core principle is to discretize the time derivative using a weighted average of solutions at two consecutive time steps ($t_n$ and $t_{n+1}$, where the time step $\Delta t = t_{n+1} - t_n$). For a temporal PDE with linear term $\mathcal{L}u = \nu \nabla^2 u$ (where $\nabla^2$ denotes the Laplacian) and nonlinear term $\mathcal{N}(u) = -\mathbf{v} \cdot \nabla u + f$ , the CN scheme yields:

$$\frac{u^{n+1} - u^n}{\Delta t} = \frac{1}{2} \left( \mathcal{L}u^{n+1} + \mathcal{N}(u^{n+1}) \right) + \frac{1}{2} \left( \mathcal{L}u^n + \mathcal{N}(u^n) \right), \tag{2}$$

Since the nonlinear term at the $t_{n+1}$ step is difficult to handle directly, a semi-implicit approach is adopted: $\mathcal{N}(u^{n+1})$ is replaced with $\mathcal{N}(u^n)$. Rearranging the equation ultimately gives:

$$u^{n+1} = \left( \left( I + \frac{\mathcal{L}}{2} \Delta t \right) u^n + \mathcal{N}(u^n) \right) (I - \frac{\mathcal{L}}{2} \Delta t)^{-1}, \tag{3}$$

Here, $u^n$ and $u^{n+1}$ are solutions at $t_n$ and $t_{n+1}$; the nonlinear term $\mathcal{N}(u)$ uses $u^n$ (known from the previous step) to avoid complex coupling. Rearranging and solving the equation (often in the frequency domain, where $\nabla^2$ becomes lap) gives $u^{n+1}$. To build the dataset, start from the initial condition $u^0$, iterate this process for $T$ time steps, and collect the spatiotemporal solution pairs $\{(t_n, u^n, f^n)\}_{n=0}^T$. For 2D equations, spatial discretization uses a Fourier pseudospectral method, and time advancement follows the Crank-Nicolson (CN) scheme. At each time step, the linear diffusion operator is diagonal in spectral space, allowing the linear system to be solved via efficient pointwise operations with complexity $O(N_{dof})$. The computational bottleneck lies in evaluating the nonlinear convective term, which requires Fast Fourier Transforms (FFTs) to toggle between physical and spectral spaces, carrying a complexity of $O(N_{dof} \log N_{dof})$. Here, $N_{dof} = S_x S_y$ represents the total spatial degrees of freedom. Consequently, the overall time complexity for generating $N_{sample}$ independent samples (each with $T$ time steps) is $O(N_{sample} T N_{dof} \log N_{dof})$.

## 3.2 NONLIEAR TEMPORAL DATASETS

Nonlinear temporal partial differential equations (PDEs) are fundamental mathematical tools for describing complex spatiotemporal dynamic systems across science and engineering (e.g., fluid flow, heat transfer, and chemical reactions). Their general form is typically expressed as:

$$\frac{\partial u(\boldsymbol{x}, t)}{\partial t} = \mathcal{L}(u(\boldsymbol{x}, t)) + \mathcal{N}(u(\boldsymbol{x}, t)) + f(\boldsymbol{x}, t) \qquad \boldsymbol{x} \in \Omega, \qquad (4)$$

$$u(\boldsymbol{x}, t) = g(\boldsymbol{x}, t) \qquad \boldsymbol{x} \in \partial\Omega, \qquad (5)$$

$$\nabla u(\boldsymbol{x}, t) \cdot \mathbf{n} = h(\boldsymbol{x}, t) \qquad \boldsymbol{x} \in \partial\Omega, \qquad (6)$$

where: $\boldsymbol{x} = (x_1, x_2, ..., x_d) \in \Omega$ denotes the $d$-dimensional spatial coordinate (with $\Omega$ as the spatial domain, e.g., $\Omega = [0, 1]^2$ for 2D problems), $t \in [0, T]$ is the time variable (with $T$ as the total time horizon), and $u(\boldsymbol{x}, t)$ is the unknown solution field (e.g., velocity for fluid flow, temperature for heat transfer) that varies with both space and time. $\mathcal{L}(\cdot)$ represents the linear spatial operator (e.g., $\mathcal{L}(u) = \nu\nabla^2 u$, where $\nu$ is the diffusion coefficient and $\nabla^2$ is the Laplacian operator describing linear diffusion). $\mathcal{N}(\cdot)$ is the nonlinear operator that introduces complexity (e.g., $\mathcal{N}(u) = -\mathbf{u} \cdot \nabla u$ for convection in Navier-Stokes equations, where the product of the solution $\mathbf{u}$ and its spatial gradient $\nabla u$ leads to nonlinearity). Finally, $f(\boldsymbol{x}, t)$ is the external source/sink term (e.g., a heat source in thermal PDEs) that drives or modifies the system's evolution. To ensure the uniqueness of the solution, boundary conditions are imposed on the domain boundary $\partial\Omega$: common types include Dirichlet conditions (specifying $u(\boldsymbol{x}, t) = g(\mathbf{x}, t)$ for $\boldsymbol{x} \in \partial\Omega$, where $g$ is a given function) and Neumann conditions (specifying $\nabla u(\boldsymbol{x}, t) \cdot \mathbf{n} = h(\boldsymbol{x}, t)$ for $\boldsymbol{x} \in \partial\Omega$, where $\mathbf{n}$ is the outward unit normal vector of $\partial\Omega$ and $h$ is a given function).

## 4 METHOD

In existing algorithms, we typically generate initial conditions and RHS of nonlinear temporal PDEs from a specific distribution, then feed them into traditional solvers for such PDEs to obtain the solution function $u(\boldsymbol{x})$. However, taking the semi-implicit method as mentioned in 3.1 as an example, we have to iterate over thousands of time steps to ensure solution stability —this demands significant computational resources and time. Yet in practice, datasets used to train neural operators typically only require the solution at a few to a dozen time steps. Thus, a natural question emerges: can we reduce the number of iterations to accelerate dataset generation?

Thus, unlike traditional methods, HOPSS acts directly on training-level data, thereby avoiding iteration over a large number of time steps. The specific process is as follows: we first generate a set of solutions using a high-precision solver, then align them to the time steps required for training via downsampling—these serve as the base solutions in our solution space. Next, the base functions are fed into the generator, where two solutions are randomly selected: one is scaled by a small scalar to act as the homologous perturbation term, and then combined with the other to generate a new solution, along with small random noise. Finally, based on these new solutions and their corresponding physical equations, we compute the right-hand side (RHS) of the equation; these collectively form new solution pairs. This strategy acts directly on training-level data, with the only exception being the generation of base solutions. Thus, when generating new data, it avoids the need to iterate over a large number of time steps.

Figure 1: Overview of the HOPSS method. First, generating a series of solutions using the existing method and downsampling as base solutions. Then, randomly select two base solutions and apply the homologous perturbation to generate new solutions. Finally, calculating the RHS using the new solutions.

The HOPSS method primarily comprises three key steps:

1. **Base solution generation**: In this phase, we generate high-precision solution functions using a high-precision solver and format them to meet the requirements for model training.

2. **Homologous perturbation in solution space**: In this stage, we apply homologous perturbation to the base solutions to generate a large number of new candidate solutions.

3. **Computation of equation RHS**: In this phase, we calculate the corresponding right-hand side (RHS) of the physical equation based on the generated candidate solutions, ensuring they satisfy the constraints of the target physical equation.

## 4.1 Solution Functions Generation

The HOPSS method generates a set of base functions by first producing solutions based on a designated distribution consistent with real-world physical scenarios—these base functions typically number 100 to 500, denoted herein as $N_{base}$. For the generation of these solutions, this paper adopts existing high-precision methods. Subsequently, we downsample the generated solutions to align them with the time steps required for training, reducing the number of time steps from several thousand to several dozen. These processed solutions serve as the base functions of HOPSS and constitute the foundational elements of its solution space.

## 4.2 Homologous Perturbation in Solution Space

The homologous perturbation process entails introducing small-scale functional perturbations to existing base solution functions. Specifically, we randomly select two base solution functions: one acts as the primary function $u_i$, and the other as the homologous perturbation term, denoted $u_j$. To prevent the generated new solution function from exhibiting excessive fluctuations, we multiply the perturbation term $u_j$ by a small constant $\mu$ (typically $\mu \approx 10^{-3}$). After combining these two functions, we further add small-scale time-invariant random noise $\xi$—specifically Gaussian noise where $\xi \sim \mathbb{N}(0, \varsigma I)$ with $\varsigma \approx 10^{-4}$ and high-frequency truncation —to enhance the robustness and diversity of the dataset. The mathematical expression for this process is:

$$u_{\text{new}} = u_i + \mu \cdot u_j + \xi. \tag{7}$$

## 4.3 Compute Variation of RHS

To satisfy the constraints of the physical equations, we recompute the variation of the right-hand side (RHS) at the training data level. Using the same discretization method as employed earlier, we derive a system of equations corresponding to the target physical equation. We then substitute the

newly generated $u_{\text{new}}$ and the primary function $u_i$ into this discretized system of equations—this enables us to calculate the variation $\delta f$ of the original RHS term $f_i$, and further obtain the new RHS term $f_{\text{new}}$. Subsequently, $u_{\text{new}}$ and $f_{\text{new}}$ form new solution pairs, which serve as the new dataset.

Taking a general equation with a source term as an example (see Eq. 4) and a specific example case is provided in Appendix E:

we begin with a base solution pair $(u_i, f_i)$ that satisfies the equation:

$$\frac{\partial u_i}{\partial t} = \mathcal{L}(u_i) + \mathcal{N}(u_i) + f_i \tag{8}$$

The new solution is generated via homologous perturbation: $u_{\text{new}} = u_i + v$, where $v = \mu u_j + \xi$. We now require that $u_{\text{new}}$ satisfies the PDE with a new source term $f_{\text{new}}$:

$$\frac{\partial u_{\text{new}}}{\partial t} = \mathcal{L}(u_{\text{new}}) + \mathcal{N}(u_{\text{new}}) + f_{\text{new}} \tag{9}$$

Substituting $u_{\text{new}} = u_i + v$ into the left-hand side of the above equation and using the linearity of the derivative and the linear operator $\mathcal{L}$, we get:

$$\frac{\partial u_i}{\partial t} + \frac{\partial v}{\partial t} = \mathcal{L}(u_i) + \mathcal{L}(v) + \mathcal{N}(u_i + v) + f_{\text{new}} \tag{10}$$

Now, we subtract the equation satisfied by the base solution $u_i$:

$$\left(\frac{\partial u_i}{\partial t} + \frac{\partial v}{\partial t}\right) - \frac{\partial u_i}{\partial t} = (\mathcal{L}(u_i) + \mathcal{L}(v) + \mathcal{N}(u_i + v) + f_{\text{new}}) \tag{11}$$

$$- (\mathcal{L}(u_i) + \mathcal{N}(u_i) + f_i) \tag{12}$$

Simplifying, we obtain the general expression for $f_{\text{new}}$:

$$f_{\text{new}} = f_i + \underbrace{\frac{\partial v}{\partial t} - \mathcal{L}(v) - [\mathcal{N}(u_i + v) - \mathcal{N}(u_i)]}_{\delta f} \tag{13}$$

## 5 THEORETICAL ANALYSIS

### 5.1 EXISTING METHOD

Solving temporal PDEs inherently involves spatiotemporal discretization, where the primary computational expense stems from iterative time step evolution and the associated solution of spatially discretized linear systems LeVeque (2007); Hughes (2000). Conventional numerical methods for temporal PDEs typically combine temporal discretization (e.g., 4th-order Runge-Kutta (RK4) Hairer et al. (1993), Crank-Nicolson (CN) scheme Crank & Nicolson) and spatial discretization (e.g., Finite Element Method (FEM) Hughes (2000), Finite Difference Method (FDM) LeVeque (2007)); the computational complexity of these methods is determined by the interplay of spatial grid scale, time step count, and linear system solving cost.

Notably, for the semi-implicit scheme, the cost of time iterations dominates the overall computational process: for the traditional method, with $N_{dof}$ as the total number of degrees of freedom in space, this step exhibits a time complexity of $O(N_{dof} \log N_{dof})$ Dong et al. (2024), and a linear system must be solved at each time step to update the PDE solution. Given $T$ total time steps, usually exceeding thousands, the overall time complexity of the entire solving process amounts to $O(T N_{dof} \log N_{dof})$. If $N_{sample}$ datapoints are generated, the all-time complexity comes up to $O(N_{sample} T N_{dof} \log N_{dof})$.

### 5.2 OUR HOPSS METHOD

According to the introduction referred to in Section 4, our method consists of three steps: base solution generation, variational action, and RHS computation.

In the first step, we use traditional methods to generate $N_b$ base solutions. According to the analysis above, its time complexity is $O(N_b T N_{dof} \log N_{dof})$. Since the cost of the downsampling step is far

lower than that of data generation, we reasonably ignore it. We then operate on training-level data, which typically has smaller $T'$ (time steps) and $N'_{dof}$ (degrees of freedom in space) than the $T$ and $N_{dof}$ used in traditional data generation. Typically, $T'$ is $\frac{T}{1000}$ or smaller.

In the subsequent two steps, the time complexity is $O(N_{\text{new}}T'N'_{dof}\log N'_{dof})$. Since $T'$ is much smaller than $T$ and can be approximated as a constant, the time complexity for subsequent new data generation is approximately $O(N_{\text{new}}N'_{dof}\log N'_{dof})$—which is one order of magnitude lower than that of traditional methods.

Thus, the total time complexity of the HOPSS method is $O(N_b T N_{dof}\log N_{dof} + N_{\text{new}}N'_{dof}\log N'_{dof}) \approx O(N_b T N_{dof}\log N_{dof})$. Consequently, the theoretical acceleration ratio of HOPSS depends on the gap between $N_b$ (the number of base solutions) and $N_{sample}$ (the number of samples required for generating an equivalent dataset via traditional methods). For more details, please refer to Appendix G.

## 6 EXPERIMENT

In this chapter, we compare our proposed data generation method with existing data generation methods.

### 6.1 EXPERIMENT SETUP

Our analysis focuses on two key performance indicators, both critical for evaluating the effectiveness of data generation methods:

- Time cost of data generation
- Test loss of neural operator models trained on the generated data

In our experiments, we test two widely recognized and adopted neural operator models—among the most prominent and prevalent in data-driven PDE algorithms:

- FNO (Fourier Neural Operator Li et al. (2021))
- Transolver( Wu et al. (2024))

We also evaluate three types of PDE problems with significant applications in science and engineering:

- Navier-Stokes equations( Li et al. (2021))
- Burgers' equation( Xie et al. (2013))
- KdV equation (Korteweg-de Vries equation Shen (1993))

**Baselines.** The primary time cost of existing data generation methods stems from iterating over thousands of time steps, despite only dozens of time steps being required for model training. We use the traditional methods to generate solution functions, which serve as our baselines. The details of the experiment are shown in Appendix B.1 and parameters of the generated dataset in Appendix B.3.

### 6.2 MAIN RESULT

Our main results across all datasets and models are presented in Table 1. Further details and hyperparameters are provided in the Appendix B.2. Based on these results, we draw the following observations.

Firstly, the HOPSS method exhibits significant acceleration compared to traditional methods—particularly for the Navier-Stokes equations, where the acceleration ratio reaches 10 times. The time cost of our method consists of three components: generating basis functions, performing homologous perturbation, and computing the RHS. Generating basis functions requires iterating over thousands of time steps, which are then downsampled to dozens of time steps. The other stages

Table 1: Performance comparison between traditional methods and HOPSS on different PDE problems. The first column lists the method used to generate the dataset and the number of training instances. The first row represents the corresponding PDE problem and the training models.

| Method | Navier-Stokes | | | KdV | | | Burgers | | |
|---|---|---|---|---|---|---|---|---|---|
| | TIME(s) | FNO | Transolver | TIME(s) | FNO | Transolver | TIME(s) | FNO | Transolver |
| Tradition1000 | 1.20e4 | 6.7e−3 | 1.4e−2 | 2.16e3 | 8.7e−3 | 3.1e−2 | 4.29e4 | 6.1e−2 | 5.9e−2 |
| HOPSS1000 | 1.22e3 | 7.6e−3 | 1.5e−2 | 1.08e3 | 1.7e−2 | 3.6e−2 | 2.15e4 | 6.2e−2 | 7.2e−2 |
| HOPSS10000 | 1.38e3 | 3.3e−3 | 8.8e−3 | 1.09e3 | 9.8e−3 | 3.8e−2 | 2.16e4 | 4.7e−2 | 7.3e−2 |

involve operating on data with dozens of time steps. Thus, the primary time cost of the entire process lies in generating the basis functions. This implies that our HOPSS method can generate large volumes of training data at low cost, underscoring the efficiency of our approach.

Secondly, across different PDE problems and with various neural operators, datasets generated by the HOPSS method exhibit comparable performance to those from traditional methods. For example, in the case of the Navier-Stokes equations, Transolver and FNO achieve nearly identical performance when trained on datasets of the same size. Moreover, as dataset sizes increase, our method can even outperform traditional methods. This demonstrates that the HOPSS method simultaneously accelerates dataset generation while maintaining approximate precision.

### 6.3 Hyperparaments Analysis

Table 2: Burgers equation hyperparameter test results. Left: Influence of perturbation level $\mu$ (with fixed Gaussian noise and $N_b = 100$). Middle: Influence of the number of base solutions $N_b$ (with fixed Gaussian noise and $\mu = 10^{-3}$). Right: Influence of noise type (with fixed $\mu = 10^{-3}$ and $N_b = 500$). Test loss is obtained from FNO.

| Perturbation Level | Test Loss | | $N_b$ | Test Loss | | Noise Type | Test Loss |
|---|---|---|---|---|---|---|---|
| $10^{-1}$ | 0.182 | | 100 | 0.0783 | | Gaussian | 0.0621 |
| $10^{-2}$ | 0.0787 | | 200 | 0.0652 | | Perlin | 0.0625 |
| $10^{-3}$ | 0.0783 | | 300 | 0.0650 | | Multi_sine | 0.0624 |
| $10^{-4}$ | 0.0784 | | 400 | 0.0639 | | Random_walk | 0.0622 |
| $10^{-5}$ | 0.0790 | | 500 | 0.0621 | | Zero | 0.0628 |

This section will show how to influence the dataset performance for the different hyperparameters.

**Perturbation level** $\mu$: As illustrated in the first subtable, dataset performance degrades as the perturbation level increases. In this experiment, we fixed the noise type (Gaussian noise) and used 100 basis solutions for both the Burgers equation and the FNO model. When the perturbation level is set to 0.1, the test loss reaches 0.182, indicating significant performance degradation. In contrast, reducing the perturbation level to 0.001 lowers the test loss to 0.0783, a substantial improvement. This suggests that excessively large perturbations may cause the generated data to become more likely out-of-distribution, thereby undermining the model's ability to learn meaningful patterns.

**Number of base solutions** $N_b$: Consistent with the results in the second subtable, dataset quality exhibits a strong correlation with the number of base solutions. In this experiment, we fixed the noise type (Gaussian noise) and set the perturbation level at 0.001—with tests conducted on both the Burgers equation and the FNO model. As the number of base solutions increases from 100 to 500, the test loss decreases from 0.0783 to 0.0621, resulting in a performance improvement of approximately 20%. This trend indicates that a larger number of base solutions helps construct a more complete subspace of the solution space, thereby enhancing the quality of the dataset.

**Noise type**: From the third subtable, it is evident that noise type has a negligible impact on model performance. The details of the noise are shown in Appendix D. In this experiment, we fixed the number of basis solutions at 500 and set the perturbation level to 0.001—consistent across both the

Burgers equation setup and the FNO model. We evaluated five experimental scenarios: four distinct noise types (Gaussian, Perlin, Multi-sine, and Random-walk noise) and a no-noise condition, and the resulting test loss ranged from 0.621 to 0.628, no more than 5%. This consistency underscores that our method demonstrates strong robustness to variations in noise type.

### 6.4 TIME STEPS INFLUENCE ON TRADITIONAL METHOD

In this section, we analyze the influence of different time steps on the traditional method, focusing specifically on two key metrics: dataset generation time cost and the performance of the subsequently trained models. As shown in Table 3, time steps exert a significant impact on both generation time cost and model training performance. For time cost: as the time step decreases from $5 \times 10^{-3}$ to $1 \times 10^{-3}$, the generation time cost increases nearly 9-fold from 1317.47 s to 11980.42 s. In terms of model performance, the FNO test loss decreases substantially from 0.106 to 0.0067, representing a marked improvement in prediction accuracy. This observation reveals a clear trade-off: smaller time steps yield better model performance but come at the cost of significantly higher dataset generation time.

Table 3: Performance comparison under different time step sizes for the Navier-Stokes problem in the traditional method

| Time Step Sizes | Time Cost (s) | Test Loss |
|---|---|---|
| $1 \times 10^{-3}$ | 11980.42 | 0.0067 |
| $2 \times 10^{-3}$ | 4380.12 | 0.024 |
| $5 \times 10^{-3}$ | 1317.47 | 0.106 |

### 6.5 ABLATION RESULT

Finally, we performed an ablation study to illustrate the critical impact of the solution generation component in our HOPSS method on dataset quality. As shown in Table 4, we evaluated datasets generated via the Mixup method and our HOPSS method (for detailed implementation of Mixup, refer to Appendix C). Experimental results confirm the effectiveness of HOPSS in high-quality dataset generation: when models are trained on Mixup-generated data, the test errors for FNO and Transolver reach 0.312 and 1.53, respectively—values that indicate the Mixup-generated dataset offers minimal training value. In contrast, our HOPSS method proves effective in generating datasets efficiently while ensuring accurate model predictions.

Table 4: Ablation experiment results comparing datasets generated using different methods

| Method | FNO | Transolver |
|---|---|---|
| Mixup | 0.312 | 1.53 |
| HOPSS | 0.0076 | 0.015 |

### 6.6 SUPPLEMENTARY EXPERIMENTS

To comprehensively evaluate the robustness and generality of the HOPSS method, we conducted six additional experiments detailed in Appendix F. These experiments systematically investigate critical aspects of HOPSS, including its sensitivity to hyperparameters, performance scalability across diverse equation types, and comparative efficacy against alternative data generation techniques. Key findings are summarized below, highlighting the method's consistency under varied conditions.

**Impact of Basis Function Selection F.1:** We evaluated HOPSS under multiple random seeds (42, 3407, 9999) to rigorously assess its stability against stochastic initialization. Results indicate negligible error fluctuations—for instance, test loss variations on Burgers' equation remain within a narrow margin of ±0.001 across seeds. This confirms that HOPSS is highly robust to randomness in basis function selection, ensuring reproducible outcomes in practical applications.

**Influence of Temporal Resolution F.2:** By varying time steps (10, 16, 20), we examined how temporal discretization granularity affects HOPSS. The method achieves superior accuracy with finer resolutions (e.g., test loss of 1.7e-2 at 20 steps with 10,000 samples), consistently outperforming traditional methods while reducing computational costs by approximately an order of magnitude. This demonstrates HOPSS's capability to efficiently capture fine-grained temporal dynamics without sacrificing precision.

**Validation on Additional Equations F.3:** Extended tests on Fitzhugh-Nagumo (FN Manafian (2010)) and Klein-Gordon (KG Denton (2024)) equations further validate HOPSS's generality beyond fluid dynamics. On KG equations, HOPSS achieves a test loss of 9.7e-3 with 10,000 samples, significantly lower than traditional methods (2.0e-2 with 1,000 samples). This underscores its adaptability to diverse nonlinear temporal PDEs, including oscillatory and reaction-diffusion systems.

**Comparison with LPSDA Method F.4:** HOPSS consistently surpasses the LPSDA Brandstetter et al. (2022) method in error reduction across benchmarks. For example, on Burgers' equation with 10,000 samples, HOPSS attains a test loss of 4.7e-2, compared to LPSDA's 5.7e-2. A hybrid LPSDA+HOPSS strategy yields only marginal gains (test loss of 5.6e-2), emphasizing that HOPSS's standalone perturbation mechanism is sufficient for high-quality data synthesis.

**Data Quality via t-SNE and Spectral Analysis F.5:** Qualitative analysis using t-SNE van der Maaten & Hinton (2008) visualization and frequency-domain spectral decomposition confirms that HOPSS-generated data preserve the topological distribution and key spectral characteristics of original solutions. In contrast, LPSDA introduces noticeable artifacts, such as irregular clustering in low-dimensional embeddings, indicating potential distribution shifts that may impair model generalization.

**Effect of Solution Function Count F.6:** Scaling the number of base solutions ($N_b$) from 100 to 500 monotonically improves HOPSS performance. Even with a modest $N_b = 100$, HOPSS outperforms traditional methods (e.g., test loss of 7.4e-2 vs. 2.0e-1). Optimal results are achieved at $N_b = 500$ (test loss of 4.5e-2 with 10,000 samples), demonstrating that a moderate base set size enables efficient data expansion without prohibitive computational overhead.

**Robustness across different test sets F.7:** We assessed HOPSS's generalization by training on datasets generated with varying random seeds and evaluating on distinct test sets. we tested the model trained on HOPSS-generated data (10,000 samples from 500 base solutions) across test sets with random seeds 0, 42, 3407, and 10086, yielding test errors of 4.6e-2, 4.7e-2, 4.7e-2, and 4.5e-2, respectively, demonstrating minimal variation.

**Residual Analysis F.8:** We conducted a detailed residual analysis to further validate the physical fidelity of HOPSS-generated datasets. Residuals were 1.23e-4 for traditional (1,000 samples), 1.37e-4 for HOPSS (1,000 samples), and 1.37e-4 for HOPSS (10,000 samples), confirming high fidelity.

Collectively, these experiments validate HOPSS as a robust and efficient approach for generating high-quality training data, with consistent performance across varied settings, including different equations, hyperparameters, and baseline comparisons.

## 7 CONCLUSION AND FUTURE WORK

**Conclusion**: In this paper, we propose the HOPSS method for generating datasets tailored to nonlinear temporal PDEs. Specifically, the method comprises three key steps: base solution generation, homologous perturbation in the solution space, and computation of the equation's right-hand side. By acting directly on the dozens of time steps required for training and computing the corresponding RHS, the HOPSS method accelerates the data generation process while preserving the precision of the data used for model training. This approach effectively overcomes a major barrier in dataset generation for nonlinear temporal PDEs.

**Future Work** : Building on HOPSS's promising performance in accelerating nonlinear temporal PDE dataset generation, we will advance the work via two targeted directions: First, we will optimize base solution selection to enhance solution space quality. Specifically, we will introduce solution space coverage analysis and active learning: by quantifying initial base solution representativeness, we will develop a framework to filter optimal ones. This prioritizes filling coverage gaps, boosting base solution diversity, and physical comprehensiveness to enhance perturbation and dataset generalization. Second, we will develop a unified physics-informed metric for dataset quality, moving beyond over-reliance on indirect indicators like model test loss. This integrated metric will encompass dual dimensions: physical consistency (e.g., adherence to PDE conservation laws, error against high-precision references) and data utility (e.g., sample diversity, cross-PDE generalization capacity). These will be integrated into a unified pipeline for comprehensive dataset assessment to guide HOPSS optimization.

## ETHICS STATEMENT

We have manually reevaluated the dataset we created to ensure it is free of any potential for discrimination, human rights violations, bias, exploitation, and any other ethical concerns.

## REPRODUCIBILITY STATEMENT

To ensure the reproducibility of our findings, all source code and datasets used in our experiments are included in the supplementary material. The provided materials are sufficient to replicate the main results presented in this paper.

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

# A  USAGE OF LLMS

Throughout the preparation of this manuscript, Large Language Models (LLMs) were utilized as a writing and editing tool. Specifically, we employed LLMs to improve the clarity and readability of the text, refine sentence structures, and correct grammatical errors. All final content, including the core scientific claims, experimental design, and conclusions, was conceived and written by us, and we take full responsibility for the final version of this paper.

# B  SPECIFIC EXPERIMENTAL DETAILS

## B.1  HARDWARE SETUP

The data generation is running on Intel(R) Xeon(R) Silver 4316, and the models are training on a GeForce RTX 4090 GPU with 24GB of memory.

## B.2  MODEL SET

**FNO1d**: we employ 3 FNO layers with learning rate 0.0001, batch size 20, epochs 1000, modes 16, and width 20.

**FNO2d**: we employ 4 FNO layers with learning rate 0.001, batch size 60, epochs 500, modes30, and width 60.

**Transolver_Irregular_Mesh**: we set 3 hidden layers with 64 dimensions for each hidden layer, batch size 4, heads 8, slice nums 32, learning rate 0.002, and epochs 1000.

**Transolver_Structured_Mesh_2D**: we set 3 hidden layers with 64 dimensions for each hidden layer, batch size 8, heads 4, slice nums 32, learning rate 0.001, and epochs 500.

## B.3  DATA

### B.3.1  NAVIER-STOKES EQUATIONS

In this research, we delve into two-dimensional Navier-Stokes equations for a viscous, incompressible fluid in vorticity form on the unit torus, which are governed by the equation(Li et al. (2021)).

$$
\begin{cases}
\partial_t w(x,t) + u(x,t) \cdot \nabla w(x,t) = \nu \Delta w(x,t) + f(x,t), & x \in (0,1)^2,\ t \in (0,T] \\
\nabla \cdot u(x,t) = 0, & x \in (0,1)^2,\ t \in [0,T] \\
w(x,0) = w_0(x), & x \in (0,1)^2
\end{cases}
\tag{14}
$$

where $u \in C([0,T]; H_{\text{per}}^r((0,1)^2; \mathbb{R}^2))$ for any $r > 0$ is the velocity field, $w = \nabla \times u$ is the vorticity, $w_0 \in L_{\text{per}}^2((0,1)^2; \mathbb{R})$ is the initial vorticity, $\nu \in \mathbb{R}_+$ is the viscosity coefficient, and $f \in L_{\text{per}}^2((0,1)^2; \mathbb{R})$ is the forcing function. For our experimental setup, the dataset is generated by the Crank-Nicolson method. We generate the dataset with $gridsize = 128$, $T = 10$, $\nu = 1e^{-4}$ and $\Delta t = 1e^{-3}$. Then we downsample the dataset to the $gridsize = 64$ and $\Delta t = 0.5$ for training the model. The force function is generated from a Gaussian Random Field (GRF) methodology, with a time constant $\tau = 2.0$ and a decay exponent $\alpha = 2.5$. In the HOPSS method, we use 100 solution functions as basis functions.

### B.3.2  BURGERS EQUATIONS

In this research, we investigate one-dimensional Burgers equations, expressed as (Xie et al. (2013)):

$$
\frac{\partial u(x,t)}{\partial t} + u(x,t) \frac{\partial u(x,t)}{\partial x} - \frac{1}{R} \frac{\partial^2 u(x,t)}{\partial x^2} = f(x,t).
\tag{15}
$$

Here, $u(x,t)$ denotes the velocity field; $R$ is the Reynolds number, a dimensionless parameter characterizing the ratio of inertial forces to viscous forces; and $f(x,t)$ represents the time-dependent forcing function. For our experimental setup, $R$ is set to 1000, indicating a flow regime with relatively strong inertial effects, while the forcing function $f(x,t)$ is generated using a Gaussian

Random Field (GRF) method with key parameters: $\tau = 7$ (time scale, controlling the temporal correlation of the random field), $\alpha = 2.5$ (decay exponent, governing the spatial correlation decay rate), and $\sigma = 7^2$ (variance, determining the amplitude of random fluctuations). For numerical discretization, the original spatial grid size (number of spatial sampling points) is 1024 with a time step $\Delta t = 5 \times 10^{-3}$ (temporal resolution), which we downsample to a grid size of 64 and $\Delta t = 5 \times 10^{-2}$ for model training, and in the HOPSS method, 500 solution functions are adopted as basis functions.

### B.3.3 FORCED KDV EQUATIONS

We also study one-dimensional forced Korteweg-de-Vries (KdV) equations, expressed as (Shen (1993)):

$$u_t + \lambda u_x + 2\alpha u u_x + \beta u_{xxx} = f'(x), \quad -\infty < x < \infty \tag{16}$$

Where $u(x,t)$ is the wave amplitude; $\lambda$ is the linear advection coefficient; $\alpha$ is the nonlinear coefficient (governing wave steepening); $\beta$ is the dispersion coefficient (counteracting steepening via wave dispersion); and $f'(x)$ denotes the spatial derivative of the forcing function $f(x)$. For our experimental setup, coefficients are set as $\alpha = -0.5$, $\beta = -1.0$, and $\lambda = 0$, configuring the equation to describe weakly nonlinear dispersive waves, while $f'(x)$ is directly generated via GRF with parameters: $\tau = 5.0$ (time scale), $\alpha = 2.5$ (spatial decay exponent), and $\sigma = 1.0$ (variance of the random field). For numerical discretization, the original spatial grid size is 512 with 10000 time steps (total temporal sampling points), which are downsampled to a grid size of 64 and 20 time steps for model training, and in the HOPSS method, 500 solution functions are used as basis functions.

### B.4 FITZHUGH–NAGUMO EQUATIONS

The Fitzhugh-Nagumo (FN) equation is a nonlinear temporal partial differential equation that models excitable media, such as neuronal dynamics. It combines diffusion with a cubic nonlinearity and is defined as:

$$u_t = D u_{xx} + u(u-a)(1-u) + f(x,t), \quad 0 < a < 1, \tag{17}$$

where $u(x,t)$ is the state variable (e.g., membrane potential), $D$ is the diffusion coefficient, $a$ is a parameter controlling the excitation threshold, and $f(x,t)$ is a spatiotemporal forcing term. The equation exhibits rich dynamics, including wave propagation and pattern formation, making it a benchmark for testing neural operators. For our experimental setup, coefficients are set as $D = 1e-3$ and $a = 0.10$. And $f$ is directly generated via GRF with parameters : $\tau = 7$ and $\alpha = 2.5$. For numerical discretization, the original spatial grid size is 1024 with 10000 time steps (total temporal sampling points), which are downsampled to a grid size of 64 and 10 time steps for model training, and in the HOPSS method, 500 solution functions are used as basis functions.

### B.5 KLEIN-GORDON EQUATIONS

The Klein-Gordon equation is a relativistic wave equation that arises in quantum field theory and nonlinear wave propagation. In our study, we consider its nonlinear form expressed as:

$$u_{tt} = u_{xx} + u^3 + f(x,t) \tag{18}$$

where the cubic nonlinear term $u^3$ introduces rich physical phenomena including soliton solutions and wave self-interactions. In our experimental setup, the external forcing term $f$ is generated using a Gaussian Random Field (GRF) with parameters $\tau = 7$ and $\alpha = 2.5$. For numerical discretization, the original spatial grid size is set to 1024 with 10,000 time steps. These data are subsequently downsampled to a grid size of 64 and 20 time steps for model training. In the HOPSS method, 500 solution functions are adopted as basis functions.

## C MIXUP METHOD

The mathematical equation can be expressed as :

$$u_{new} = \sum_{i=0}^{N_b} \alpha_i u_i, \sum_{i=0}^{N_b} \alpha_i = 1 \tag{19}$$

where $u_i$ is the solutions and $\alpha_i$ is a constant. In our experiment, $\alpha_i$ is sampled from a standard Gaussian distribution and then normalized. $N_b$ is set to 100.

## D  NOISY INTRODUCTION

We corrupt clean data $x$ (a vector or the last dimension of a tensor) with several synthetic noise models. A unified relative amplitude parameter $\varepsilon = \texttt{noise\_level}$ is mapped to an absolute scale

$$A = \varepsilon \cdot \max_i |x_i|, \qquad A > 0 \text{ (fallback } A = 10^{-8} \text{ if } \max_i |x_i| = 0). \tag{20}$$

All raw noise patterns $\tilde{\eta}$ are rescaled to $\eta = A\,\tilde{\eta}/\max|\tilde{\eta}|$ (except the Gaussian case which is sampled directly with variance $A^2$). The noisy signal is $x' = x + \eta$.

### D.1  GAUSSIAN NOISE

Standard i.i.d. zero-mean Gaussian (baseline):

$$\eta_i^{(\mathrm{G})} \sim \mathbb{N}(0, A^2), \quad x_i' = x_i + \eta_i^{(\mathrm{G})}. \tag{21}$$

Parameters used: only $\varepsilon$ (e.g. $\varepsilon = 10^{-3}$). No spatial correlation; flat spectrum.

### D.2  MULTI-SINE NOISE

A smooth low-frequency superposition of $K$ sinusoidal modes with random coefficients and phases over normalized coordinate $s \in [0, 1]$ discretized into $L$ points:

$$\tilde{\eta}(s) = \sum_{k=1}^{K} \Big[ a_k \sin(2\pi k s + \phi_k) + b_k \cos(2\pi k s + \psi_k) \Big], \tag{22}$$

With $a_k, b_k \sim \mathcal{U}(-1, 1)$ and phases $\phi_k, \psi_k$ sampled uniformly (implemented as a single phase per mode). Normalization:

$$\eta^{(\mathrm{MS})}(s) = A \frac{\tilde{\eta}(s)}{\max_s |\tilde{\eta}(s)|}.$$

Parameters: $K = \texttt{multi\_sine\_k} = 8$; relative amplitude $\varepsilon$. Result: band-limited (low-frequency) non-Gaussian smooth perturbation.

### D.3  PERLIN NOISE

One-dimensional Perlin procedural noise with $C$ lattice cells ($C + 1$ gradient points). Let $s \in [0, 1]$, $t = sC$, $i = \lfloor t \rfloor$, $u = t - i \in [0, 1)$. Random gradients $G_i \sim \mathcal{U}(-1, 1)$. Use the quintic fade

$$\mathrm{fade}(u) = 6u^5 - 15u^4 + 10u^3.$$

Define endpoint contributions $v_0 = G_i u$, $v_1 = G_{i+1}(u - 1)$ and interpolate:

$$\tilde{\eta}(s) = v_0 + (v_1 - v_0)\,\mathrm{fade}(u), \qquad \eta^{(\mathrm{P})}(s) = A \frac{\tilde{\eta}(s)}{\max_s |\tilde{\eta}(s)|}.$$

Parameters: $C = \min(\texttt{perlin\_cells}, L - 1)$ with $\texttt{perlin\_cells} = 32$; relative amplitude $\varepsilon$. Produces locally smooth, multi-scale, non-Gaussian structure.

### D.4  RANDOM WALK NOISE

Cumulative uniform increments (strong correlation, non-stationary before centering):

$$\delta_\ell \sim \mathcal{U}(-1, 1), \quad \tilde{\eta}_\ell = \sum_{j=1}^{\ell} \delta_j, \quad \bar{\eta} = \frac{1}{L}\sum_{\ell=1}^{L} \tilde{\eta}_\ell, \quad \hat{\eta}_\ell = \tilde{\eta}_\ell - \bar{\eta}, \quad \eta_\ell^{(\mathrm{RW})} = A \frac{\hat{\eta}_\ell}{\max_\ell |\hat{\eta}_\ell|}.$$

Parameters: only $\varepsilon$ (no extra hyperparameter). Produces low-frequency drift-like perturbations after mean removal.

**Broadcast over higher dimensions.**  For tensors shaped $(N, S)$ or $(N, S, T)$, the non-Gaussian patterns are generated along the last axis ($S$) and broadcast to other leading dimensions; per-sample independent noise can be obtained by generating a separate pattern per batch slice.

# E THE COMPUTATION OF RHS

The core principle for computing the new source term $f_{\text{new}}$ is to enforce that the newly generated solution $u_{\text{new}}$ satisfies the original PDE. For the general nonlinear temporal PDE given by:

$$\frac{\partial u}{\partial t} = \mathcal{L}(u) + \mathcal{N}(u) + f \tag{23}$$

we begin with a base solution pair $(u_i, f_i)$ that satisfies the equation:

$$\frac{\partial u_i}{\partial t} = \mathcal{L}(u_i) + \mathcal{N}(u_i) + f_i \tag{24}$$

The new solution is generated via homologous perturbation: $u_{\text{new}} = u_i + v$, where $v = \mu u_j + \xi$. We now require that $u_{\text{new}}$ satisfies the PDE with a new source term $f_{\text{new}}$:

$$\frac{\partial u_{\text{new}}}{\partial t} = \mathcal{L}(u_{\text{new}}) + \mathcal{N}(u_{\text{new}}) + f_{\text{new}} \tag{25}$$

Substituting $u_{\text{new}} = u_i + v$ into the left-hand side of the above equation and using the linearity of the derivative and the linear operator $\mathcal{L}$, we get:

$$\frac{\partial u_i}{\partial t} + \frac{\partial v}{\partial t} = \mathcal{L}(u_i) + \mathcal{L}(v) + \mathcal{N}(u_i + v) + f_{\text{new}} \tag{26}$$

Now, we subtract the equation satisfied by the base solution $u_i$:

$$\left( \frac{\partial u_i}{\partial t} + \frac{\partial v}{\partial t} \right) - \frac{\partial u_i}{\partial t} = (\mathcal{L}(u_i) + \mathcal{L}(v) + \mathcal{N}(u_i + v) + f_{\text{new}}) \tag{27}$$

$$- (\mathcal{L}(u_i) + \mathcal{N}(u_i) + f_i) \tag{28}$$

Simplifying, we obtain the general expression for $f_{\text{new}}$:

$$f_{\text{new}} = f_i + \underbrace{\frac{\partial v}{\partial t} - \mathcal{L}(v) - [\mathcal{N}(u_i + v) - \mathcal{N}(u_i)]}_{\delta f} \tag{29}$$

This formula is general. The term $\mathcal{N}(u_i+v) - \mathcal{N}(u_i)$ represents the change in the nonlinear operator due to the perturbation $v$. Crucially, this term is computed exactly based on the defined nonlinear operator $\mathcal{N}$; it does not involve a linearization approximation. For any specific PDE (like Burgers, Navier-Stokes, etc.), one simply substitutes the corresponding operators $\mathcal{L}$ and $\mathcal{N}$ into the above equation. For example, in Burgers equation, $\mathcal{N}(u) = -uu_x$, and substituting this into the above equation yields the specific result shown in Eq. 37 of the paper.

## E.1 BURGERS EQUATIONS

$$\frac{\partial u}{\partial t} + uu_x = \nu u_{xx} + f(x, t), \tag{30}$$

We assume both $(u_{\text{new}}, f_{\text{new}})$ and $(u_i, f_i)$ satisfy Eq. 30.

$$\frac{\partial u_i}{\partial t} + u_i u_{i,x} = \nu u_{i,xx} + f_i \tag{31}$$

$$\frac{\partial u_{\text{new}}}{\partial t} + u_{\text{new}} u_{\text{new},x} = \nu u_{\text{new},xx} + f_{\text{new}} \tag{32}$$

According to the generation rule for $u_{\text{new}}$, we introduce the transformation:

$$u_i = u_{\text{new}} - v, \tag{33}$$

where $v = \mu u_j + \xi$ is a newly defined composite variable, with $\mu$ being a constant and $\xi = \xi(x)$ being a spatially dependent function.

Now, we substract Eq. 31 from Eq. 32:

$$\frac{\partial u_{\text{new}}}{\partial t} - \frac{\partial u_i}{\partial t} + u_{\text{new}}u_{\text{new},x} - u_i u_{i,x} = \nu(u_{\text{new},xx} - u_{i,xx}) + f_{\text{new}} - f_i \tag{34}$$

Substituting $u_i = u_{\text{new}} - v$ into the above equation, we have:

$$\frac{\partial u_{\text{new}}}{\partial t} - \frac{\partial(u_{\text{new}} - v)}{\partial t} + u_{\text{new}}u_{\text{new},x} - (u_{\text{new}} - v)(u_{\text{new},x} - v_x) = \nu(u_{\text{new},xx} - (u_{\text{new},xx} - v_{xx})) + f_{\text{new}} - f_i \tag{35}$$

Simplifying, we obtain:

$$\frac{\partial v}{\partial t} + u_{\text{new}}u_{\text{new},x} - u_{\text{new}}u_{\text{new},x} + u_{\text{new}}v_x + v u_{\text{new},x} - v v_x = \nu v_{xx} + f_{\text{new}} - f_i \tag{36}$$

Rearranging terms, we derive the expression for $f_{\text{new}}$:

$$f_{\text{new}} = f_i + \underbrace{\frac{\partial v}{\partial t} + (u_{new}v)_x - v v_x - \nu v_{xx}}_{\delta f}. \tag{37}$$

# F  SUPPLEMENTARY EXPERIMENT

## F.1  THE IMPACT OF BASIS FUNCTION SELECTION ON HOPSS PERFORMANCE

Table 5: Error Comparison of HOPSS Method Under Different Random Seeds

| Method | Random Seed 42 | Random Seed 3407 | Random Seed 9999 |
| --- | --- | --- | --- |
| Tradition 1000 | 6.1e-2 | 6.1e-2 | 6.1e-2 |
| HOPSS 1000 | 6.1e-2 | 6.2e-2 | 6.3e-2 |
| HOPSS 10000 | 4.7e-2 | 4.6e-2 | 4.6e-2 |

To verify the stability of the HOPSS method with respect to the randomness of basis function selection, we fixed the number of randomly selected basis functions to 500 in the Burgers' equation. We set three different random seeds (42, 3407, 9999) and generated 1000 and 10000 samples respectively using the HOPSS method. The errors were tested on the FNO1d model. The experimental results are shown in Table 5. It can be seen that the error fluctuations of the HOPSS method under different random seeds are small, indicating that it has good stability against the randomness of basis function selection.

## F.2  THE IMPACT OF TEMPORAL RESOLUTION ON HOPSS PERFORMANCE

To analyze the impact of temporal resolution on the performance of the HOPSS method, we set the time steps to 10, 16, and 20 respectively in Burgers' equation-based experiments. We then evaluated the prediction errors of various methods under the FNO1d model, where the model takes forward $\frac{t}{2}$ data as input and predicts backward $\frac{t}{2}$ data. The experimental results are presented in Table 6. As the time step increases , the errors of all methods exhibit a decreasing trend. Furthermore, the HOPSS method demonstrates a more prominent advantage under high temporal resolution (specifically, time step = 20): for 10,000 samples, its error is as low as $1.7 \times 10^{-2}$.

## F.3  EXTENDED VALIDATION OF HOPSS ON MULTIPLE EQUATIONS

To verify the generality of the HOPSS method, we extended it to the Fitzhugh-Nagumo (FN) and Klein-Gordon (KG) equations and tested its errors in the FNO model. The experimental results are shown in Table 7: For the FN equation, the error of the HOPSS method is close to that of the Tradition method (3.2e-2). For the KG equation, the HOPSS method shows a significant advantage, with an error of only 9.7e-3 for 10000 samples, which is much lower than the 2.0e-2 of the Tradition method with 1000 samples. This indicates that the HOPSS method has better performance in complex equations.

Table 6: Error Comparison of Various Methods Under Different Time Steps

| Method | Time Step 10 | Time Step 16 | Time Step 20 |
|---|---|---|---|
| Tradition 500 | 7.5e-2 | 6.3e-2 | 5.7e-2 |
| Tradition 1000 | 6.1e-2 | 4.4e-2 | 3.8e-2 |
| HOPSS 1000 | 6.1e-2 | 4.7e-2 | 4.4e-2 |
| HOPSS 10000 | 4.7e-2 | 2.1e-2 | 1.7e-2 |

Table 7: Error Comparison of HOPSS Method on FN and KG Equations

| Method | FN Equation Error | KG Equation Error |
|---|---|---|
| Tradition 500 | 3.4e-2 | 5.0e-2 |
| Tradition 1000 | 3.2e-2 | 2.0e-2 |
| HOPSS 1000 | 3.2e-2 | 1.9e-2 |
| HOPSS 10000 | 3.2e-2 | 9.7e-3 |

## F.4 PERFORMANCE COMPARISON BETWEEN HOPSS AND LPSDA METHODS

To further verify the superiority of the HOPSS method, we conducted a comparative study between HOPSS and LPSDA methods under the Burgers' equation scenario, and additionally tested the performance of their combination strategy (LPSDA+HOPSS). It should be noted that all methods were based on an initial 500 samples: the "Enhanced 1000 " condition refers to expanding 500 samples to 1000 via either LPSDA or HOPSS, while "Enhanced 10000" follows the same logic; the LPSDA+HOPSS strategy specifically involves two-step expansion: first expanding 500 samples to 1000 via LPSDA, then further expanding to 10000 via HOPSS.

Table 8: Performance Comparison of HOPSS, LPSDA, and Their Combination

| Condition | FNO Error Value | |
|---|---|---|
| | LPSDA | HOPSS |
| Tradition 500 | 7.5e-2 | |
| Tradition 1000 | 6.1e-2 | |
| Enhanced 1000 | 7.0e-2 | 6.1e-2 |
| Enhanced 10000 | 5.7e-2 | 4.7e-2 |
| LPSDA+HOPSS | 5.6-2 | |

Experimental results are presented in Table 8: The error of the HOPSS method (6.1e-2 when enhanced to 1000 samples, 4.7e-2 when enhanced to 10000 samples) is significantly lower than that of the LPSDA method (7.0e-2 when enhanced to 1000 samples, 5.7e-2 when enhanced to 10000 samples). Although the error of the LPSDA+HOPSS combination strategy (5.6e-2 ) is slightly better than that of LPSDA alone (5.7e-2 at 10000 samples), it remains higher than the error of HOPSS alone at 10000 samples. This result confirms that the core advantages of the HOPSS method are irreplaceable.

## F.5 VALIDATION OF SYNTHETIC DATA BASED ON t-SNE DIMENSIONALITY REDUCTION AND SPECTRUM ANALYSIS

To further demonstrate the superiority of HOPSS-generated synthetic data over LPSDA, we conducted a comparative analysis from the dual perspectives of data distribution and frequency-domain characteristics. Employing t-SNE dimensionality reduction and spectral analysis, we evaluated the similarity between synthetic data (from both methods) and the original data. As illustrated in Figure 2, the key findings are as follows:

- t-SNE Visualization: While both HOPSS and LPSDA methods effectively generated data that align with the distribution of data produced by traditional methods, a notable difference emerged: LPSDA exhibited significant point aggregation at the edges of the distribution, a phenomenon absent in both HOPSS-generated data and the traditional method's output. This suggests that HOPSS produces data with a more natural and consistent distribution compared to LPSDA.

- Spectral Analysis: The main frequency components of HOPSS-synthesized data aligned more closely with those of the original data than did LPSDA-synthesized data. This confirms HOPSS's superior ability to replicate the authentic frequency-domain characteristics of the original data. Furthermore, HOPSS maintained excellent physical consistency of the data, indicating that the generated data adheres to the underlying physical constraints of the original system.

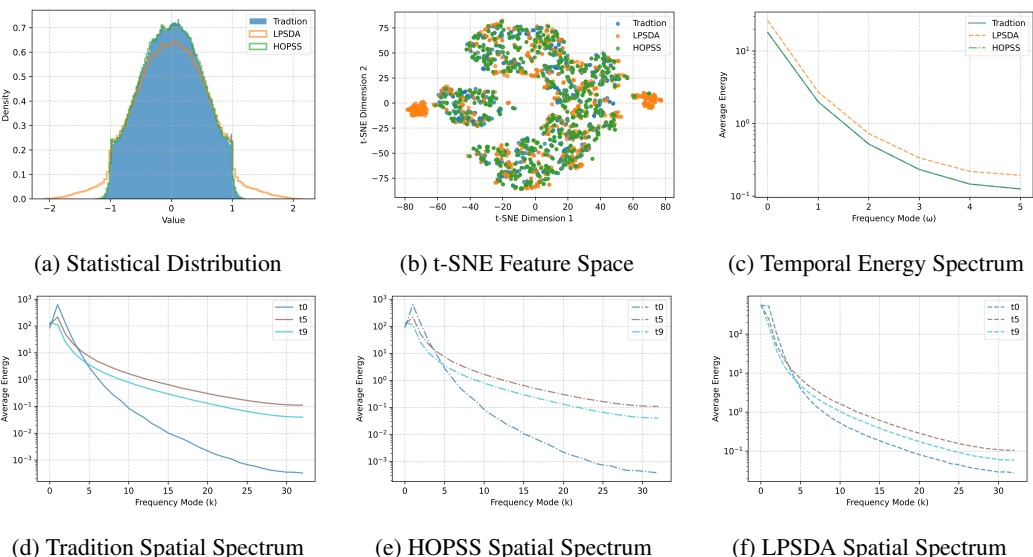

(a) Statistical Distribution  (b) t-SNE Feature Space  (c) Temporal Energy Spectrum

(d) Tradition Spatial Spectrum  (e) HOPSS Spatial Spectrum  (f) LPSDA Spatial Spectrum

Figure 2: Comparison of Synthetic Data Quality: HOPSS vs. LPSDA vs. Tradition: (a) compares the statistical distributions of the datasets generated by the three methods; (b) presents the feature space distributions of these datasets after t-SNE dimensionality reduction; (c) displays the power spectra of the datasets obtained via Fourier transform in the time domain; (d), (e), and (f) respectively illustrate the spectral analyses via Fourier transform in the spatial domain of the three datasets at three specific time points: t=0, t=5, and t=9.

### F.6  THE IMPACT OF THE NUMBER OF SOLUTION FUNCTIONS ON HOPSS PERFORMANCE

To determine the optimal number of solution functions, we tested the errors of the Tradition and HOPSS methods under different $N_b$ values (100, 200, 500). The experimental results are shown in Table 9: For the Tradition method, as $N_b$ increases, the error decreases significantly (2.0e-1 when $N_b$=100, 8.0e-2 when $N_b$=500). For the HOPSS method, the performance is optimal when $N_b$=500 (6.2e-2 for 1000 samples, 4.5e-2 for 10000 samples). Moreover, under the same $N_b$, the error of the HOPSS method is much lower than that of the Tradition method, verifying that it has a lower dependence on the number of solution functions and better performance.

Table 9: Error Comparison of Various Methods Under Different Numbers of Solution Functions

| Training Method | $N_b$=100 | $N_b$=200 | $N_b$=500 |
|---|---|---|---|
| Tradition ($N_b$ samples) | 2.0e-1 | 1.3e-1 | 8.0e-2 |
| Tradition (1000 samples) | 6.1e-2 | 6.1e-2 | 6.1e-2 |
| HOPSS (1000 samples) | 7.4e-2 | 6.5e-2 | 6.2e-2 |
| HOPSS (10000 samples) | 7.2e-2 | 5.6e-2 | 4.5e-2 |

### F.7  ROBUSTNESS ACROSS DIFFERENT TEST SETS

To assess the robustness of HOPSS-generated models, we trained an FNO model on 10,000 samples generated from 500 base solutions via HOPSS method using the Burgers equation. We then evalu-

ated the model on four different test sets, each generated with distinct random seeds (0, 42, 3407, 10086). The test errors are summarized in Table 10, showing minimal variation and confirming the stability of HOPSS.

Table 10: Test Errors on Different Test Sets for HOPSS-Trained FNO Model

| Random Seed | Test Error |
|---|---|
| 0 | 4.6e-2 |
| 42 | 4.7e-2 |
| 3407 | 4.7e-2 |
| 10086 | 4.5e-2 |

## F.8 PDE RESIDUAL ANALYSIS

We computed the PDE residual for datasets generated by different methods on the Burgers equation, defined as the average of the difference between the left-hand side and the right-hand side. we performed piecewise cubic Hermite interpolation (PCHIP) on the datasets generated by the two methods in both the time and spatial domains, interpolating the spatial grid size to 1024 and the time step to 200. Subsequently, we adopted the same discrete format as the numerical solver to calculate the PDE residual at each time step.

Table 11: PDE Residuals for Different Dataset Generation Methods

| Method | Residual |
|---|---|
| Tradition (1000 samples) | 1.23e-4 |
| HOPSS (1000 samples) | 1.37e-4 |
| HOPSS (10000 samples) | 1.37e-4 |

Table 12: PDE Residuals at Different Time Steps for Tradition and HOPSS

| Time Step ($T$) | Tradition | HOPSS |
|---|---|---|
| $T = 2$ | 1.6e-5 | 5.4e-6 |
| $T = 5$ | 1.5e-5 | 5.3e-6 |
| $T = 8$ | 1.3e-5 | 4.4e-6 |

The results in Table 11 show that HOPSS maintains residuals comparable to the traditional method. Meanwhile, Table 12 indicates that at the same time points, HOPSS yields significantly lower PDE residuals than the traditional method. This preliminarily confirms that HOPSS effectively suppresses numerical error accumulation by recalculating the RHS after homotopy perturbation. Furthermore, HOPSS maintains good numerical stability over increasing time steps without numerical breakdown.

## F.9 LARGER PERTURBATION ANALYSIS

**Experimental Settings** To demonstrate that HOPSS remains mathematically valid under large perturbations, we conducted an experiment on the Burgers' equation. We first generated $N_b = 500$ base solutions using the standard Fourier pseudospectral method with the Crank-Nicolson scheme. Using these base solutions, we applied HOPSS to generate 1,000 new samples under varying perturbation coefficients $\mu \in \{0.1, 0.5, 1.0\}$. We recorded the generation time (Cost) and calculated the physical inconsistency (Residual) using the same formulation as defined in Appendix F.8.

**Experimental Results** The computational cost and mean residuals for generating 1,000 samples are presented in Table 13.

Table 13: Performance of HOPSS with large perturbation coefficients ($\mu$) for generating 1,000 samples.

| Perturbation $\mu$ | Generation Cost (s) | Mean Residual |
|---|---|---|
| 0.1 | 0.69 | $1.6 \times 10^{-5}$ |
| 0.5 | 0.70 | $2.1 \times 10^{-5}$ |
| 1.0 | 0.61 | $3.1 \times 10^{-5}$ |

**Analysis and Conclusion**   As shown in Table 13, the generation cost remains consistently low (less than 1 second for 1,000 samples) and is independent of the perturbation magnitude. More importantly, the residuals remain at a negligible level ($O(10^{-5})$) even when $\mu$ increases to 1.0. This confirms that HOPSS is not limited to small perturbation approximations; by explicitly calculating the corresponding source term $f_{\text{new}}$, the method ensures that the generated large-perturbation data strictly satisfies the governing physical equations.

### F.10   VARIABLE COEFFICIENT BURGERS' EQUATION

We extend the HOPSS method to the variable coefficient Burgers' equation, demonstrating its capability to handle equations with heterogeneous parameters. In this setting, we simultaneously perturb the coefficient $a(x)$ and the solution function $u(x)$, and then recompute the corresponding source term $f$ to generate valid solution tuples $(a, u, f)$. The governing equation is defined as:

$$\frac{\partial u}{\partial t} + u\frac{\partial u}{\partial x} = \frac{\partial}{\partial x}\left(a(x)\frac{\partial u}{\partial x}\right) + f(t, x), \quad x \in [0, 1], t \in [0, 1] \tag{38}$$

**Experimental Settings**   The dataset was generated using a high-precision solver employing the IMEX SSP RK3 scheme for temporal integration and the Finite Difference Method (FDM) for spatial discretization. The resolution was set to $N_x = 1024$ and $N_t = 2000$. The coefficient $a(x)$, initial condition $u_0$, and source term $f$ were sampled from Gaussian Random Fields (GRF). For the HOPSS method, we applied Gaussian noise with high-frequency truncation to the perturbations to ensure both randomness and smoothness ($\mu = 10^{-3}$).

**Performance Comparison**   We compared the traditional method against HOPSS. For the traditional approach, we generated 1,000 samples directly. For HOPSS, we first generated 500 base solutions and then expanded them to datasets of 1,000 and 10,000 samples, respectively. The computational cost (total generation time in seconds) and the mean physical residual (calculated as per Appendix F.8) are reported in Table 14.

Table 14: Comparison of computational cost and residuals between Traditional method and HOPSS for the variable coefficient Burgers' equation.

| Method | Dataset Size | Cost (s) | Mean Residual |
|---|---|---|---|
| Traditional | 1,000 | $6.22 \times 10^3$ | $4.6 \times 10^{-5}$ |
| HOPSS | 1,000 | $3.21 \times 10^3$ | $5.1 \times 10^{-5}$ |
| HOPSS | 10,000 | $3.22 \times 10^3$ | $5.2 \times 10^{-5}$ |

**Analysis and Conclusion**   The results highlight two critical advantages of HOPSS:

- **Accuracy Maintenance:** The residuals of the datasets generated by HOPSS ($5.1 \sim 5.2 \times 10^{-5}$) are comparable to those of the traditional high-precision solver ($4.6 \times 10^{-5}$). This confirms that even when perturbing both the coefficient $a(x)$ and the solution $u$, HOPSS maintains strict physical consistency.
- **Extreme Efficiency in Scaling:** Generating 1,000 samples via HOPSS cost roughly half the time of the traditional method, primarily due to the sunk cost of generating the 500 base solutions. However, the true power of HOPSS is revealed in the 10,000-sample case. Increasing the dataset size from 1,000 to 10,000 samples increased the total cost by a negligible amount (from $3.21 \times 10^3$s to $3.22 \times 10^3$s). This indicates that the marginal cost of generating new samples via HOPSS is virtually zero compared to the traditional solver, making it ideal for large-scale dataset construction.

## G   TIME COMPLEXITY ANALYSIS

We take the 2D Navier-Stokes equation for a viscous, incompressible fluid in vorticity form on the unit torus $\mathbb{T}^2 = [0, 1] \times [0, 1]$ as an example:

$$\frac{\partial \omega}{\partial t} + \mathbf{u} \cdot \nabla \omega = \nu \Delta \omega + f \tag{39}$$

where $\omega = \frac{\partial v}{\partial x} - \frac{\partial u}{\partial y}$ is the vorticity, $\mathbf{u} = (u, v)^T = \left( \frac{\partial \psi}{\partial y}, -\frac{\partial \psi}{\partial x} \right)^T$ is the velocity vector, $\psi$ is the streamfunction, $\mathbf{u} \cdot \nabla \omega$ (i.e., $u\frac{\partial \omega}{\partial x} + v\frac{\partial \omega}{\partial y}$) is the nonlinear convective term, $\Delta = \frac{\partial^2}{\partial x^2} + \frac{\partial^2}{\partial y^2}$ is the 2D Laplacian, and $f$ is the source term.

## G.1 TRADITIONAL METHOD

We first discretize spatial variables via a Fourier pseudospectral method. Let $\mathcal{F}$ denote the 2D Fourier transform, and $\hat{\omega} = \mathcal{F}(\omega)$ be the Fourier spectral coefficient of $\omega$. Applying $\mathcal{F}$ to both sides of equation 39 gives the spectral-space equation:

$$\frac{\partial \hat{\omega}}{\partial t} + \widetilde{\mathcal{N}}(\hat{\omega}) = \widetilde{\mathcal{L}}\hat{\omega} + \hat{f} \tag{40}$$

Here, $\widetilde{\mathcal{N}}(\hat{\omega}) = \mathcal{F}(\mathbf{u} \cdot \nabla \omega)$ is the nonlinear convective term in spectral space, $\hat{f} = \mathcal{F}(f)$ is the transformed source term, and $\widetilde{\mathcal{L}}$ is the linear viscous diffusion operator defined as $\widetilde{\mathcal{L}}\hat{\omega} = \nu\widetilde{\Delta}\hat{\omega}$. $\widetilde{\Delta} = -(k_x^2 + k_y^2)$ is the Fourier-transformed Laplacian, with $k_x, k_y$ being the wave numbers in the $x$ and $y$ directions.

For temporal discretization, we adopt the standard Crank-Nicolson (CN) scheme—consistently treating all terms with the average of $t^n$ and $t^{n+1}$ values :

$$\frac{\hat{\omega}^{n+1} - \hat{\omega}^n}{\Delta t} + \frac{1}{2}\left( \widetilde{\mathcal{N}}(\hat{\omega}^n) + \widetilde{\mathcal{N}}(\hat{\omega}^{n+1}) \right) = \frac{1}{2}\left( \widetilde{\mathcal{L}}\hat{\omega}^n + \widetilde{\mathcal{L}}\hat{\omega}^{n+1} \right) + \frac{1}{2}\left( \hat{f}^n + \hat{f}^{n+1} \right) \tag{41}$$

where $\Delta t$ is the time step, $\hat{\omega}^n = \hat{\omega}(t^n = n\Delta t)$, $\hat{\omega}^{n+1} = \hat{\omega}(t^{n+1} = (n+1)\Delta t)$ denote the spectral coefficients at consecutive time steps, and $\hat{f}^n = \mathcal{F}(f(t^n))$, $\hat{f}^{n+1} = \mathcal{F}(f(t^{n+1}))$ are the transformed source terms at corresponding times.

To avoid solving a nonlinear system , we apply an explicit approximation to the nonlinear term—retaining only its $t^n$ time step value: $\widetilde{\mathcal{N}}(\hat{\omega}^{n+1}) \approx \widetilde{\mathcal{N}}(\hat{\omega}^n)$. For the source term, we similarly use its $t^n$ value $\hat{f}^n$ for simplicity. Substituting $\widetilde{\mathcal{L}}\hat{\omega} = \nu\widetilde{\Delta}\hat{\omega}$ and the approximations into equation 41, we rewrite the equation as:

$$\hat{\omega}^{n+1} - \hat{\omega}^n + \Delta t \cdot \widetilde{\mathcal{N}}(\hat{\omega}^n) - \Delta t \cdot \hat{f}^n = \frac{\nu\Delta t}{2}\widetilde{\Delta}\left( \hat{\omega}^n + \hat{\omega}^{n+1} \right) \tag{42}$$

The Fourier pseudospectral method offers a distinct computational advantage due to the diagonal structure of the linear operator $\widetilde{\mathcal{L}}$ in spectral space. Since the Fourier basis functions are eigenfunctions of the Laplacian, the transformed operator $\widetilde{\Delta}$ acts as a diagonal matrix with entries corresponding to the wave numbers $(k_x, k_y)$. Consequently, the linear system decouples, allowing us to rearrange equation 42 using the identity operator $I$:

$$\left( I - \frac{\nu\Delta t}{2}\widetilde{\Delta} \right)\hat{\omega}^{n+1} = \left( I + \frac{\nu\Delta t}{2}\widetilde{\Delta} \right)\hat{\omega}^n - \Delta t \cdot \widetilde{\mathcal{N}}(\hat{\omega}^n) + \Delta t \cdot \hat{f}^n \tag{43}$$

Due to this decoupling, inverting the operator $\left( I - \frac{\nu\Delta t}{2}\widetilde{\Delta} \right)$ simplifies to a scalar division for each independent spectral coefficient. Thus, the solution $\hat{\omega}^{n+1}$ is efficiently obtained via pointwise operations:

$$\hat{\omega}^{n+1} = \frac{1 + \frac{\nu\Delta t}{2}\widetilde{\Delta}}{1 - \frac{\nu\Delta t}{2}\widetilde{\Delta}}\hat{\omega}^n + \frac{\Delta t}{1 - \frac{\nu\Delta t}{2}\widetilde{\Delta}}\left( \hat{f}^n - \widetilde{\mathcal{N}}(\hat{\omega}^n) \right) \tag{44}$$

Here, the fractions denote element-wise operations using the scalar values $1 - \frac{\nu\Delta t}{2}\widetilde{\Delta}(k_x, k_y)$ associated with each wave number.

Based on the explicit update formula equation 44, we analyze the computational cost per time step, which consists of two main parts: the linear update and the evaluation of the nonlinear term.

**Linear Operator Inversion** As derived in equation 43, the implicit linear operator $\left( I - \frac{\nu\Delta t}{2}\widetilde{\Delta} \right)$ is diagonal. Its inversion corresponds to a pointwise scalar division for each wave number. For a grid with total degrees of freedom $N_{\text{dof}} = S_x S_y$, this operation requires $O(N_{\text{dof}})$ floating-point operations.

**Nonlinear Term Evaluation** The nonlinear term $\widetilde{\mathcal{N}}(\hat{\omega}^n) = \mathcal{F}(\mathbf{u} \cdot \nabla \omega)$ is computationally the most demanding component. In the pseudospectral approach, convolution sums in spectral space are avoided to prevent $O(N_{\text{dof}}^2)$ complexity. Instead, the term is evaluated using the "transform method":

- **Differentiation:** Velocity $\hat{\mathbf{u}}$ and gradients $\widehat{\nabla \omega}$ are computed in spectral space via pointwise multiplication ($O(N_{\text{dof}})$).

- **Inverse FFT:** These quantities are transformed to physical space using the Inverse Fast Fourier Transform (IFFT). For 2D fields, this requires $O(N_{\text{dof}} \log N_{\text{dof}})$ operations.

- **Product:** The convective product $\mathbf{u} \cdot \nabla \omega$ is computed point-wise in physical space ($O(N_{\text{dof}})$).

- **Forward FFT:** The result is transformed back to spectral space using the Forward FFT ($O(N_{\text{dof}} \log N_{\text{dof}})$).

Summing these components, the dominant complexity per time step is dictated by the FFT operations:

$$\mathcal{C}_{\text{step}} = O(N_{\text{dof}} \log N_{\text{dof}}) \tag{45}$$

Therefore, for a simulation generating $N_{\text{sample}}$ samples, each evolving over $T$ time steps, the total computational complexity is:

$$\mathcal{C}_{\text{total}} = O(N_{\text{sample}} \cdot T \cdot S_x S_y \log(S_x S_y)) \tag{46}$$

### G.2 HOPSS Method

The HOPSS method decouples the data generation process into two distinct phases: (1) the generation of high-precision base solutions, and (2) the generation of new samples via homologous perturbation in the solution space. We analyze the complexity of each phase below.

**Phase 1: Base Solution Generation** We first generate $N_b$ base solutions using the traditional high-precision solver. Let $N_{\text{dof}}$ and $T$ denote the spatial degrees of freedom and time steps of the high-precision solver, respectively. As analyzed in the Theoretical Analysis section, the cost is dominated by the FFT operations required for the nonlinear term evaluation in the semi-implicit scheme. Thus, the complexity for this phase is:

$$\mathcal{C}_{\text{base}} = O(N_b \cdot T \cdot N_{\text{dof}} \log N_{\text{dof}}) \tag{47}$$

**Phase 2: Homologous Perturbation and RHS Computation** This phase generates $N_{\text{new}}$ new samples on a target training grid with $N_{\text{dof}}'$ spatial degrees of freedom and $T'$ time steps (typically $N_{\text{dof}}' \leq N_{\text{dof}}$ and $T' \ll T$). The core operation is computing the new source term $f_{\text{new}}$ for the perturbed solution $u_{\text{new}} = u_i + v$, where $v = \mu u_j + \xi$. The variational form is given by:

$$f_{\text{new}} = f_i + \underbrace{\frac{\partial v}{\partial t} - \mathcal{L}(v) - [\mathcal{N}(u_i + v) - \mathcal{N}(u_i)]}_{\delta f} \tag{48}$$

We break down the computational cost of evaluating $\delta f$ per time step on the training grid:

- **Perturbation Construction ($v$):** The linear combination $v = \mu u_j + \xi$ involves simple element-wise addition and scalar multiplication in physical space. **Cost:** $O(N_{\text{dof}}')$.

- **Time Derivative ($\partial v / \partial t$):** Consistent with the Crank-Nicolson scheme employed in the traditional solver, the temporal derivative is approximated via finite differences on the temporal grid. This involves element-wise subtraction and division operations in physical space. **Cost:** $O(N_{\text{dof}}')$.

- **Linear Operator ($\mathcal{L}(v)$):** We compute $\mathcal{L}(v)$ using the Fourier pseudospectral method. Since the linear operator is diagonal in spectral space, the computation involves:
  1. FFT of $v$ to spectral space: $O(N_{\text{dof}}' \log N_{\text{dof}}')$.
  2. Element-wise multiplication by the diagonal operator $\widetilde{\mathcal{L}}$: $O(N_{\text{dof}}')$.

3. IFFT back to physical space: $O(N'_{\text{dof}} \log N'_{\text{dof}})$.

**Cost:** $O(N'_{\text{dof}} \log N'_{\text{dof}})$.

- **Nonlinear Variation ($\mathcal{N}(u_i + v) - \mathcal{N}(u_i)$):** Similar to the traditional solver, we use the pseudospectral transform method to evaluate the convective terms. This requires computing gradients in spectral space, performing inverse FFTs, computing pointwise products in physical space, and (if necessary) transforming back. The cost is dominated by the FFTs. **Cost:** $O(N'_{\text{dof}} \log N'_{\text{dof}})$.

Summing these components, the complexity of generating one new sample pair $(u_{\text{new}}, f_{\text{new}})$ is dominated by the FFT operations, scaling as $O(T' \cdot N'_{\text{dof}} \log N'_{\text{dof}})$. Therefore, the total complexity for Phase 2 is:

$$\mathcal{C}_{\text{expansion}} = O(N_{\text{new}} \cdot T' \cdot N'_{\text{dof}} \log N'_{\text{dof}}) \tag{49}$$

**Total Complexity and Speedup**    The total computational complexity of HOPSS is the sum of both phases. Given that the number of time steps on the training grid is significantly smaller than that of the high-precision solver ($T' \ll T$), the computational cost of the expansion phase is negligible compared to the base solution generation. Thus, the total complexity can be approximated as:

$$\mathcal{C}_{\text{HOPSS}} = O(N_b \cdot T \cdot N_{\text{dof}} \log N_{\text{dof}}) + O(N_{\text{new}} \cdot T' \cdot N'_{\text{dof}} \log N'_{\text{dof}}) \approx O(N_b \cdot T \cdot N_{\text{dof}} \log N_{\text{dof}}) \tag{50}$$

Consequently, compared to the traditional method which requires simulating $N_{\text{sample}}$ samples over $T$ time steps, the theoretical acceleration ratio of HOPSS is determined by the reduction in high-precision simulations, yielding a speedup factor roughly proportional to $N_{\text{sample}}/N_b$.

