# OpenReview forum: "Accelerating Data Generation for Nonlinear temporal PDEs via homologous perturbation in solution space"
_ICLR.cc/2026/Conference — ICLR 2026 Conference Desk Rejected Submission_

### Official Review · Reviewer_EjcR · 2025-10-30

**Soundness:** 2
**Presentation:** 2
**Contribution:** 2
**Rating:** 4
**Confidence:** 3

**Summary:**

The paper propose a simple way to augment PDE datasets for time-depende problem where the RHS is part of the input, called homologous perturbation. Given a base dataset, it randomly combine two samples as

$$u_{new} = u_1 + a u_2 + \epsilon$$

and then compute the new RHS.

Numerical experiments show the homologous perturbation augmentation generally improves performance and generalization.

**Strengths:**

- The paper studies an important but often overlooked probably on efficient data generation for neural operators.
- The method is simple but seemingly effective.

**Weaknesses:**

- It is not very clear how the new RHS is computed, especially new non-linear term $u u_x$. Is it just approximate by the finite difference method?
- Please provides some error analysis on the non-linear term in Sec 4, and how it is related to the perturbation parameter.
- The method is not applicable for many problem where RHS is not part of the input.

Minor: please avoid abusing notations such as
- - line 166: $O(Nn)$, $N$ was the nonlinear term, $n$ was the time index.
- - line 271: $\Delta f$  (look like laplacian),

**Questions:**

It will be helpful to add some discussion on how the RHS is computed.

---

> ### Author Response · Authors · 2025-11-19
>
> We thank the reviewer for the insightful and valuable comments. We respond to your comments as follows and sincerely hope that our rebuttal could properly address your concerns. If so, we would deeply appreciate it if you could raise your score and your confidence. If not, please let us know your further concerns, and we will continue actively responding to your comments and improving our work.
>
> # Weakness 1 and question
>
> > It is not very clear how the new RHS is computed, especially new non-linear term . Is it just approximate by the finite difference method?
>
> > It will be helpful to add some discussion on how the RHS is computed.
>
> We thank the reviewer for raising this important point regarding the computation of the new right-hand side (RHS), particularly for nonlinear terms. We apologize for any lack of clarity in our original description and are happy to provide further explanation.
>
> The new right-hand side is **not merely computed using the finite difference method**. Intuitively, the approach we adopt to calculate the new RHS is inherently consistent with the method used to solve the base functions. For example, as detailed in our code, we use the Crank-Nicolson scheme and the spectral method to compute both the RHS and the nonlinear terms. Furthermore, in Section 4.3 of the paper, we have included a detailed example of RHS computation for the Burgers equation; in the revised version, we also provide a general RHS computation method in the AppendixE. In general, the specific computational process is as follows:
>
> 1. For a general nonlinear temporal PDE (Equation 4):
>    $$\frac{\partial u}{\partial t} = \mathcal{L}(u) + \mathcal{N}(u) + f$$
>    We start with a base solution pair $(u_i, f_i)$ that satisfies the equation:
>    $$\frac{\partial u_i}{\partial t} = \mathcal{L}(u_i) + \mathcal{N}(u_i) + f_i$$
>    A new solution is generated via homologous perturbation: $u_{\text{new}} = u_i + v$, where $v = \mu u_j + \xi$  ($\mu$ is the perturbation strength parameter, and $\xi$ is the noise term).
>
> 2. We now require the new solution $u_{\text{new}}$ to satisfy the PDE together with a new source term $f_{\text{new}}$:
>    $$\frac{\partial u_{\text{new}}}{\partial t} = \mathcal{L}(u_{\text{new}}) + \mathcal{N}(u_{\text{new}}) + f_{\text{new}} \quad \text{(1)}$$
>
> 3. Substitute $u_{\text{new}} = u_i + v$ into the left-hand side of Equation (1). Using the linearity of the derivative and the linear operator $\mathcal{L}$, we obtain:
>    $$\frac{\partial u_i}{\partial t} + \frac{\partial v}{\partial t} = \mathcal{L}(u_i) + \mathcal{L}(v) + \mathcal{N}(u_i + v) + f_{\text{new}}$$
>
> 4. Next, we subtract the equation satisfied by the base solution $u_i$:
>    $$\left( \frac{\partial u_i}{\partial t} + \frac{\partial v}{\partial t} \right) - \frac{\partial u_i}{\partial t} = \left( \mathcal{L}(u_i) + \mathcal{L}(v) + \mathcal{N}(u_i + v) + f_{\text{new}} \right) - \left( \mathcal{L}(u_i) + \mathcal{N}(u_i) + f_i \right)$$
>
> After simplification, we derive the general expression for $f_{\text{new}}$:
> $$f_{\text{new}} = f_i + \frac{\partial v}{\partial t} - \mathcal{L}(v) - \left[ \mathcal{N}(u_i + v) - \mathcal{N}(u_i) \right] \quad \text{(2)}$$
>
> This formula is universal.
>
> We hope this explanation clarifies our approach.

---

> > ### Author Response · Authors · 2025-11-19
> >
> > # weakness 2
> >
> > > Please provides some error analysis on the non-linear term in Sec 4, and how it is related to the perturbation parameter.
> >
> > We thank the reviewer for this insightful question. Regarding the error in the nonlinear term, we would like to clarify a crucial point: the essence of the HOPSS method is not to linearize the original equation or introduce truncation errors. Instead, it directly constructs a new, PDE-consistent solution function $u_{\text{new}}$ through "homologous perturbation." Subsequently, we precisely compute the matching right-hand side term $ f_{\text{new}}$ by substituting  $u_{\text{new}}$ back into the discretized original governing equation.
> >
> > Therefore, this process itself does not introduce additional theoretical errors into the nonlinear term. The errors involved mainly stem from two sources:
> > 1. The inherent numerical error of the high-precision solver used to generate the base functions;
> > 2. The numerical error arising from solving the discrete equation when calculating $ f_{\text{new}}$.
> >
> > The perturbation parameter $\mu$ serves to control the degree of deviation between the newly generated solution $ u_{\text{new}}$ and the base solution  $u_i$, ensuring that $u_{\text{new}}$ always lies on a physically plausible solution manifold. It is not used for error approximation. The advantage of our method lies in this "construction-back calculation" approach, which avoids the accumulated errors incurred by traditional methods when iteratively solving nonlinear terms over thousands of time steps.
> >
> > Furthermore, we conducted experiments to calculate the PDE residuals of the generated data. The results show that the data generated by our algorithm still maintains physical consistency:
> >
> > | Method                | Residual  |
> > |-----------------------|-----------|
> > | Tradition (1000 samples) | 1.23e-4 |
> > | HOPSS (1000 samples)    | 1.37e-4 |
> > | HOPSS (10000 samples)   | 1.37e-4 |

---

> > > ### Author Response · Authors · 2025-11-19
> > >
> > > # weakness 3
> > >
> > > > The method is not applicable for many problem where RHS is not part of the input.
> > >
> > > We thank the reviewer for this astute observation, which correctly identifies a key scope of our method. The HOPSS framework, as presented in this work, is indeed primarily designed for a specific yet important class of problems: **learning the mapping from a source term (RHS) to the solution of a PDE**. This is highly relevant to inverse problems and control scenarios, where the source term acts as either an unknown quantity or a control variable (e.g., estimating a heat source from temperature measurements, or designing a forcing term to achieve a desired flow pattern).
> > >
> > > We acknowledge that this formulation does not directly apply to scenarios where the RHS is fixed (e.g., zero in homogeneous equations) or an intrinsic component of the operator (e.g., initial value problems, where the focus is on the temporal evolution starting from a given initial condition). For such problems, the input to the neural operator typically consists of initial conditions or parameters, rather than the RHS.
> > >
> > > However, the core idea of HOPSS—**efficiently generating physically consistent solution pairs by perturbing a base solution manifold and enforcing compliance with the governing equation**—has the potential to be adapted. For instance, in the case of an initial value problem, one could generate pairs of initial conditions and their corresponding solution trajectories at a limited number of time steps. The "homologous perturbation" could then be applied to these solution trajectories, and the step of computing the RHS would be replaced by verifying consistency with the governing law. We consider this a promising direction for future work, as it would broaden the applicability of our approach.

---

> > > > ### Author Response · Authors · 2025-11-19
> > > >
> > > > # weakness 4
> > > >
> > > > > Minor: please avoid abusing notations such as
> > > > > - line 166: $O(Nn)$, $N$ was the nonlinear term, $n$ was the time index.
> > > > > - line 271: $\Delta f$(look like laplacian),
> > > >
> > > > We sincerely appreciate your careful correction! We have revised these notation expressions that were prone to confusion in the manuscript to ensure the standardization and readability of notation usage.

---

> > ### Comment · Reviewer_EjcR · 2025-11-19
> >
> > Thank the author for the response.
> >
> > Overall it would be better to improve the writing and provide more details in Sec 4.3, as Reviewer n4Qs and ked8 are all confused by the exact formulation.
> >
> > From the response, it is now clear that the methods depends on Crank-Nicolson scheme and the spectral method to compute both the RHS and the nonlinear terms. My concern is using numerical scheme, no matter FDM or spectral method, the method again introduce numerical error. It will be intereting to study how the error introduced here propogate.
> >
> > It would be better to do the numerical analysis comparing the two approaches:
> > 1. use the numerical solvers (Crank-Nicolson scheme and the spectral method) to directly generate the data, vs
> > 2. use the numerical solvers (Crank-Nicolson scheme and the spectral method) to perturbate the data using the proposed method.
> >
> > I will be happy to improve my review score, if it can be shown that the proposed method has a better convergence rate compared to directly using the numerical method such as Crank-Nicolson scheme and the spectral method (their convergence has been well studied).

---

> > > ### Author Response · Authors · 2025-11-22
> > >
> > > We would like to thank the reviewer for your valuable comments. As suggested, we have revised and restated Section 4.3 of the paper using a generalized equation formulation, and supplemented the detailed derivation process of the right-hand side (RHS) calculation for the Burgers equation in Appendix E to enhance the clarity and completeness of the content.
> > >
> > > Regarding the numerical error propagation issue you raised , we have conducted additional experiments to quantitatively compare the PDE residuals of the traditional data generation method and the HOPSS method at different time steps (T=2, 5, 8) during the solution process. Specifically, in the Burgers dataset, we performed piecewise cubic Hermite interpolation (PCHIP) on the datasets generated by the two methods in both the time and spatial domains, interpolating the spatial grid size to 1024 and the time step to 200. Subsequently, we adopted the same discrete format as the numerical solver to calculate the PDE residual at each time step, and finally extracted the results corresponding to the originally specified time steps (T=2, 5, 8). The experimental results are as follows:
> > >
> > > | Method | Tradition | HOPSS   |
> > > | ------ | --------- | ------- |
> > > | T = 2  | 1.6e-5    | 5.4e-6  |
> > > | T = 5  | 1.5e-5    | 5.3e-6  |
> > > | T = 8  | 1.3e-5    | 4.4e-6  |
> > >
> > > The results indicate that at the same time points, the PDE residuals of the solutions generated by the HOPSS method are significantly lower than those of the traditional method. This finding initially confirms that the HOPSS method can effectively suppress the accumulation of numerical errors by recalculating the RHS after homotopy perturbation. Furthermore, the HOPSS method maintains good numerical stability without numerical breakdown as the number of time steps increases.

---

> ### Author Response · Authors · 2025-11-26
>
> Dear Reviewer,
>
> We hope everything is going well for you. We are writing to gently follow up on our rebuttal and would be glad to provide any additional clarification if needed. Thank you very much for your time and effort.
>
> Authors

---

### Official Review · Reviewer_ked8 · 2025-10-31

**Soundness:** 1
**Presentation:** 3
**Contribution:** 1
**Rating:** 2
**Confidence:** 4

**Summary:**

Authors propose a particular data augmentation technique suitable for a particular family of nonlinear temporal PDEs with forcing term. Suppose that PDE $\mathcal{N}(u(x, t)) = f(x, t)$ defines evolution of physical field $u(x, t)$ and we want to learn a mapping $u(x, 0), f(x, t)\rightarrow u(x, t)$ for $t \in [0, T]$. Authors propose to collect dataset $u_i(x, t), i=1,\dots,N$ and augment it with additional solutions and forcing terms $\widetilde{u}_j(x, t),\widetilde{f}_j(x, t),\,j=1,\dots,M$ constructed as follows:
1. Select distinct $k\_1, k\_2\in \left(1,\dots, N\right)$
2. Select $\mu\ll 1$ and small random perturbation $\xi(x)$, $\left\|\xi\right\|\ll\left\|u_{k_1}\right\|$
3. Compute $\widetilde{u}(x, t) = u_{k_1}(x, t) + \mu u_{k_2}(x, t) + \xi(x, t)$ and $\widetilde{f}(x, t)$ (not explained how this is done in case of general nonlinearity)
4. Add new points to the dataset $\widetilde{u}(x, t), \widetilde{f}(x, t)$.

Authors propose to apply the scheme above along with a temporal coarsening to make augmentation computationally cheaper.

The approach is validated on three equations: Burgers, Navier-Stokes, KdV.

**Strengths:**

1. Data generation presents a main bottleneck for learning neural PDE solvers. The aim of research is to alleviate this problem, so it is well motivated.
2. The approach authors propose is lightweight and can be used to generate a large number of additional solutions with minimal overhead.

**Weaknesses:**

1. The method authors suggest is not completely specified.
2. Numerical experiments are not convincing.

I provide more details in the section below.

**Questions:**

**The method is not completely specified**

Authors explain how their method works in Section 4. The main confusing part is Section 4.3. where authors show how to compute forcing terms for a particular equation only. How this technique can be extended to the general case of PDE given in equations (4) - (6) is not explained. The problem is Burgers equation has a very specific form of nonlinearity, and for other equations one will need to perform linearisation explicitly. Can the authors supply a missing general description of their method?

**Numerical experiments are not convincing**

In my view authors provide little evidence that their method works as intended. To assess the quality of augmentation one needs to train a neural network with and without augmentation on the same data and compare test errors. This needs to be done for different numbers of samples in the train set.

More specifically, I suggest the following comparison protocol (number of samples below is adjustable):
1. For selected equations generate a dataset by standard means with $N=1100$ sample.
2. Split dataset on $1000$ train samples and $100$ test samples
3. For $N_{s}$ in $[100, 200, 300, \dots]$ do
   a. Select $N_s$ samples from $1000$ train samples
   b. Train neural network on these samples and record test error (on $100$ hold out samples)
   c. Enrich dataset with augmented samples; train second neural network and record test error
4. Report $E_{\text{test}}^{(s)}\pm \text{std}$ with and without augmentation.

If in the experiment above one can gain substantial improvement in test accuracy by training on augmented dataset, this provides evidence that augmentation is useful. Authors did not perform this kind of test, so it is impossible to tell whether their augmentation works or not.

From the data present one can not make any useful conclusion. For example, data from Table 1 shows that for augmented dataset accuracy drops, but only slightly. What does it imply? Maybe, it simply implies that the dataset is not challenging enough, and with $100$ samples one can already have decent accuracy.

This conclusion is partially supported by Table 2, where one can see that for small perturbation level accuracy drops but only slightly. Essentially, for perturbation $10^{-5}$ we almost have unperturbed data. This may suggest $100$ samples of unperturbed data is enough to reach mean relative test error $\simeq 8\%$.

**Other questions:**
1. What if modelled PDE does not have a source term?
2. What to do when there are other parameters, e.g., diffusion coefficient $k(x)$ in Burgers equation $\frac{\partial u(x, t)}{\partial t} + u(x, t)\frac{\partial u(x, t)}{\partial x} = \frac{\partial}{\partial x} k(x)\frac{\partial u(x, t)}{\partial x}$? How augmentation is applied in this case?
3. How the augmentation approach by author compares with other augmentation techniques for PDEs, e.g., https://arxiv.org/abs/2202.07643, https://arxiv.org/abs/2301.12730, https://arxiv.org/abs/2501.14604, etc
4. Why do authors call the experiment in Section 6.5. an ablation study? In Section 6.5. authors compare their method HOPSS with another method Mixup, that they introduce in the Appendix. Mixup clearly generates out of distribution data and is superficially related to HOPSS. Typically, ablation studies include some form of "perturbation" of the proposed approach, e.g., "we remove layers not from the architecture and report accuracy". Can the authors explain why Section 6.5. constitutes ablation study?
5. In many places when citation is added authors forgot to add whitespace characters. Examples: line 49, line 54, line 57, line 93, line 94, etc. Can the authors please correct that?
6. I find the description on lines 111-121 confusing. For evolution PDEs, the standard way to describe numerical solutions is as follows. The discretisation is first done for spatial differential operators (method of lines). After that we are left with a system of ordinary differential equations that is solved with matching techniques, e.g., RK4, trapezoidal rule (=Crank-Nicolson), etc. Besides, it is not true that nonlinear terms always require linearisation. As the authors pointed below, there are IMEX schemes.
7. In equation (3) authors divide by $(1 - \frac{L}{2}\Delta t)$. I see two problematic parts: (i) $L$ is a matrix, so in place of $1$ one should use identity matrix (the same is valid for the numerator); (ii) for inverse matrix a typical notation is $(I - \frac{L}{2}\Delta t)^{-1}$, it is not customary to represent inverse matrix as fraction.
8. Complexity specified in lines 164-167 is doubtful. Authors did explain how spatial discretisation is performed and without that it is hard to estimate the cost of matrix inversion. Can the authors provide more details how they compute computation complexity? What is the number of data points $N$? What is a dimension of spatial discretisation $n$? What is the number of physical dimensions authors consider?
9. Lines 193-194: "In existing algorithms, we typically generate initial conditions and RHS of nonlinear temporal PDEs randomly, ..." This is not the case. There is a standard statistical formulation for operator learning, see, e.g., https://arxiv.org/abs/2010.08895. We generate data from the specified distribution, because this is a setup of an operator learning problem. The distribution is chosen based on the application: it encodes a subset of interesting physical parameters.

---

> ### Author Response · Authors · 2025-11-19
>
> We thank the reviewer for the insightful and valuable comments. We respond to your comments as follows and sincerely hope that our rebuttal could properly address your concerns. If so, we would deeply appreciate it if you could raise your score and your confidence. If not, please let us know your further concerns, and we will continue actively responding to your comments and improving our work.
>
> # weakness 1
>
> > **The method is not completely specified**
> >
> >
> > Authors explain how their method works in Section 4. The main confusing part is Section 4.3. where authors show how to compute forcing terms for a particular equation only. How this technique can be extended to the general case of PDE given in equations (4)-(6) is not explained. The problem is Burgers equation has a very specific form of nonlinearity, and for other equations one will need to perform linearisation explicitly. Can the authors supply a missing general description of their method?
>
> We sincerely thank the reviewer for this extremely important and professional comment. We apologize for the lack of clarity in our original description and will revise the AppendixE to include a **general formulation** applicable to the broad class of PDEs defined by Equations (4)–(6). The technique is indeed general and does not require equation-specific linearization; the Burgers equation was only used as a concrete example for illustrative purposes. Below is the proposed general description:
>
>
> The core principle for computing the new source term $f_{\text{new}}$ is to ensure that the newly generated solution $u_{\text{new}}$ satisfies the original PDE. For a general nonlinear temporal PDE given by (Equation 4):
> $$\frac{\partial u}{\partial t} = \mathcal{L}(u) + \mathcal{N}(u) + f$$
>
> We start with a **base solution pair** $(u_i, f_i)$ that satisfies the above equation:
> $$\frac{\partial u_i}{\partial t} = \mathcal{L}(u_i) + \mathcal{N}(u_i) + f_i$$
>
> A new solution is generated via **homologous perturbation**: $u_{\text{new}} = u_i + v$, where $v = \mu u_j + \xi$ ($\mu$ denotes the perturbation strength, and $\xi$ is the noise term). We now require $u_{\text{new}}$ to satisfy the PDE with a new source term $f_{\text{new}}$:
> $$\frac{\partial u_{\text{new}}}{\partial t} = \mathcal{L}(u_{\text{new}}) + \mathcal{N}(u_{\text{new}}) + f_{\text{new}} \quad \text{(1)}$$
>
>
> Substitute $u_{\text{new}} = u_i + v$ into the left-hand side of Equation (1). Using the linearity of the derivative and the linear operator $\mathcal{L}$, we obtain:
> $$\frac{\partial u_i}{\partial t} + \frac{\partial v}{\partial t} = \mathcal{L}(u_i) + \mathcal{L}(v) + \mathcal{N}(u_i + v) + f_{\text{new}}$$
>
>
> Next, we subtract the equation satisfied by the base solution $u_i$ from both sides:
> $$\left( \frac{\partial u_i}{\partial t} + \frac{\partial v}{\partial t} \right) - \frac{\partial u_i}{\partial t} = \left( \mathcal{L}(u_i) + \mathcal{L}(v) + \mathcal{N}(u_i + v) + f_{\text{new}} \right) - \left( \mathcal{L}(u_i) + \mathcal{N}(u_i) + f_i \right)$$
>
>
> After simplification, we derive the **general expression for $f_{\text{new}}$**:
> $$f_{\text{new}} = f_i + \frac{\partial v}{\partial t} - \mathcal{L}(v) - \left[ \mathcal{N}(u_i + v) - \mathcal{N}(u_i) \right] \quad \text{(2)}$$
>
>
> This formula is universal. The term $\mathcal{N}(u_i + v) - \mathcal{N}(u_i)$ represents the change in the nonlinear operator caused by the perturbation $v$. Critically, this term is computed **exactly** based on the defined nonlinear operator $\mathcal{N}$, with no linearization approximation involved. For any specific PDE (e.g., Burgers, Navier-Stokes), one only needs to substitute the corresponding $\mathcal{L}$ (linear operator) and $\mathcal{N}$ (nonlinear operator) into Equation (2). For example, in the Burgers equation, $\mathcal{N}(u) = -u u_x$; substituting this into Equation (2) yields the specific result shown in Equation (11) of the paper. We will add this general derivation to the manuscript to ensure clarity and demonstrate the universal applicability of the RHS computation technique.

---

> > ### Author Response · Authors · 2025-11-19
> >
> > # weakness 2
> >
> > >**Numerical experiments are not convincing**
> > >
> > >In my view authors provide little evidence that their method works as intended. To assess the quality of augmentation one needs to train a neural network with and without augmentation on the same data and compare test errors. This needs to be done for different numbers of samples in the train set.
> >
> > In our work, HOPSS is primarily designed as a data generation algorithm that accelerates the creation of large-scale datasets from scratch, rather than a data augmentation technique applied to existing small datasets. Additionally, we have partially addressed the reviewer’s suggestion through Supplementary Experiment 6. In this experiment, we tested different training set sizes (e.g., $N_s = 100, 200, 500$), where $N_s$ samples were randomly selected from a fixed training set of 1000 samples. The subsequent rows labeled "HOPSS 1000" and "HOPSS 10000" represent datasets expanded from these $N_s$ samples to 1000 and 10000 samples using the HOPSS algorithm, respectively.
> >
> > The results demonstrate that using only $N_s$ samples leads to suboptimal performance (e.g., a test error of 2.0e-1 when $N_s = 100$), while datasets augmented via HOPSS achieve performance comparable to or even better than the baseline (e.g., HOPSS 10000 reduces the error to 4.5e-2 when $N_s = 500$). This confirms that HOPSS effectively improves data quality and model generalization, especially when starting with limited base samples.
> >
> >
> > | Training Method          | $N_{\text{base}} = 100$ | $N_{\text{base}} = 200$ | $N_{\text{base}} = 500$ |
> > |--------------------------|------------------------------|------------------------------|------------------------------|
> > | Tradition ($N_{\text{base}}$ samples) | 2.0e-1                       | 1.3e-1                       | 8.0e-2                       |
> > | Tradition (1000 samples) | 6.1e-2                       |         6.1e-2                     |        6.1e-2                      |
> > | HOPSS (1000 samples)     | 7.4e-2                       | 6.5e-2                       | 6.2e-2                       |
> > | HOPSS (10000 samples)    | 7.2e-2                       | 5.6e-2                       | 4.5e-2                       |

---

> > > ### Author Response · Authors · 2025-11-19
> > >
> > > # question 1
> > >
> > > > What if modelled PDE does not have a source term?
> > >
> > > We thank the reviewer for this astute observation, which correctly identifies a key scope of our method. The HOPSS framework, as presented in this work, is indeed primarily designed for a specific yet important class of problems: learning the mapping from a source term (RHS) to the solution of a PDE. This is highly relevant to inverse problems and control scenarios, where the source term serves as either an unknown quantity or a control variable (e.g., estimating a heat source from temperature measurements, or designing a forcing term to achieve a desired flow pattern).
> > >
> > > We acknowledge that this formulation does not directly apply to problems where the RHS is fixed (e.g., zero in homogeneous equations) or is an intrinsic component of the operator (e.g., initial value problems, where the focus is on the temporal evolution starting from a given initial condition). For such problems, the input to the neural operator would typically be only the initial condition or parameters.
> > >
> > > However, the core idea of HOPSS—**efficiently generating physically consistent solution pairs by perturbing a base solution manifold and enforcing compliance with the governing equation**—could potentially be adapted. For instance, in the case of an initial value problem, one could generate pairs of initial conditions and their corresponding solution trajectories at a limited number of time steps. The "homologous perturbation" could then be applied to these solution trajectories, and the step of computing the RHS would be replaced by verifying consistency with the governing law. We consider this a promising direction for future work to broaden the applicability of our approach.

---

> > > > ### Author Response · Authors · 2025-11-19
> > > >
> > > > # question 2
> > > >
> > > > > 1. What to do when there are other parameters, e.g., diffusion coefficient $k(x)$ in Burgers equation $\frac{\partial u(x,t)}{\partial t} + u(x, t)\frac{\partial u(x,t)}{\partial x} = \frac{\partial}{\partial x}k(x)\frac{\partial u(x,t)}{\partial x}$ ? How augmentation is applied in this case?
> > > >
> > > > We thank the reviewer for this important question regarding PDEs with additional parameters, such as a spatially varying diffusion coefficient $ k(x) $. The HOPSS method is designed for general nonlinear PDEs and can be extended to handle such cases seamlessly.
> > > >
> > > > In the HOPSS framework, PDE parameters (e.g., $ k(x) $) are treated as fixed coefficients that are part of the equation definition and remain unchanged during the augmentation process. The core idea is to generate new solution pairs $ (u_{\text{new}}, f_{\text{new}}) $ by applying a transformation to the base solutions, similar to the derivation in Eqs. 8–11 of our paper for the standard Burgers equation. For the given Burgers equation with $ k(x) $:
> > > >
> > > > $$\frac{\partial u(x,t)}{\partial t} + u(x,t)\frac{\partial u(x,t)}{\partial x} = \frac{\partial}{\partial x}\left(k(x)\frac{\partial u(x,t)}{\partial x}\right) + f(x,t)$$
> > > >
> > > > We first note that this equation can be written in the general form of Eq. 4: $ \frac{\partial u}{\partial t} = \mathcal{L}(u) + \mathcal{N}(u) $, where the linear operator $ \mathcal{L}(u) = \frac{\partial}{\partial x}\left(k(x)\frac{\partial u}{\partial x}\right) $and the nonlinear operator $ \mathcal{N}(u) = -u\frac{\partial u}{\partial x} $ .
> > > >
> > > >
> > > > The augmentation process follows these steps:
> > > > 1. **Base solution generation**: We generate high-precision base solutions $ u_i $ for the PDE with the specific $ k(x) $using traditional solvers.
> > > > 2. **Homologous perturbation**: We apply the transformation $ u_i = u_{\text{new}} - v $ (as in Eq. 9), where $v$ is a composite variable ( $v = \mu u_j + \xi(x) $).
> > > > 3. **Compute new RHS**: By substituting $ u_i = u_{\text{new}} - v $ into the PDE, we derive the equation for $u_{\text{new}} $. The new solution $u_{\text{new}} $ will satisfy the same PDE form but with a modified source term $f_{\text{new}}$. Specifically:
> > > >    - The left-hand side (time derivative and nonlinear term) transforms similarly to the standard case.
> > > >    - The linear operator $ \mathcal{L}(u) $ now involves $ k(x) $; when substituting, the terms with $ k(x) $ will appear in the derivation of $\delta f $.
> > > >    - The new source term $ f_{\text{new}} $ is computed as $ f_{\text{new}} = f_i + \delta f $, where $ \delta f $ now includes contributions from the variable coefficient $ k(x) $. For example, $ \delta f $ will contain terms like $ \frac{\partial}{\partial x}\left(k(x)\frac{\partial v}{\partial x}\right) $ from the linear operator part.
> > > >
> > > >
> > > > Mathematically, this yields:
> > > > $$\frac{\partial u_{\text{new}}}{\partial t} + u_{\text{new}}\frac{\partial u_{\text{new}}}{\partial x} = \frac{\partial}{\partial x}\left(k(x)\frac{\partial u_{\text{new}}}{\partial x}\right) + f_{\text{new}}$$
> > > >
> > > > where $ f_{\text{new}} = \delta f $, and $ \delta f $ is derived from the transformation and includes dependencies on $ k(x) $. This ensures that $ u_{\text{new}} $ is a valid solution for the PDE with the given $ k(x) $.
> > > >
> > > > In practice, during spatial discretization , $ k(x) $ is discretized and treated as a fixed array. The computation of $ \delta f $ follows the same discrete operators as used for the base solver. Thus, HOPSS can handle variable parameters without modification, as the method operates on the discretized system level. Our approach focuses on generating data that conforms to the PDE structure, regardless of parameter complexity.
> > > >
> > > > We acknowledge that our current experiments focused on constant parameters for simplicity, but the method is general. Future work will include explicit examples with variable coefficients.

---

> > > > > ### Author Response · Authors · 2025-11-19
> > > > >
> > > > > # question 3
> > > > >
> > > > > > How the augmentation approach by author compares with other augmentation techniques for PDEs, e.g., https://arxiv.org/abs/2202.07643, https://arxiv.org/abs/2301.12730, https://arxiv.org/abs/2501.14604, etc
> > > > >
> > > > > We thank the reviewer for the suggestion to compare HOPSS with similar data augmentation methods—this helps better contextualize its novelty and value. In response, we would like to emphasize that our document already includes a direct comparison with a relevant data augmentation approach, **Lie Point Symmetry Data Augmentation (LPSDA)**, in Supplementary Experiment 4. This experiment evaluates the performance of HOPSS and LPSDA on the Burgers equation using the FNO (Fourier Neural Operator) model.
> > > > >
> > > > > The results show that both LPSDA and HOPSS outperform the baseline method in improving model performance. However, HOPSS achieves lower error rates, demonstrating its superior effectiveness. Furthermore, a key distinction lies in their underlying mechanisms: LPSDA relies on the symmetry properties of the target PDEs, which limits its applicability to scenarios where such symmetries are known and easily identifiable. In contrast, HOPSS operates directly on the temporal resolution of the data—this design allows it to be applied to a broader range of PDE problems, even when the equations lack obvious symmetry properties or when identifying such symmetries is computationally expensive.
> > > > >
> > > > > The detailed comparison results are presented in the table below:
> > > > >
> > > > > | Method          | LPSDA     | HOPSS     |
> > > > > |-----------------|-----------|-----------|
> > > > > | Tradition 500   | 7.5e-2    | 7.5e-2    |
> > > > > | Tradition 1000  |6.1e−2        | 6.1e−2        |
> > > > > | Enhanced 1000   | 7.0e-2    | 6.1e−2    |
> > > > > | Enhanced 10000  | 5.7e-2    | 4.7e−2    |

---

> > > > > > ### Author Response · Authors · 2025-11-19
> > > > > >
> > > > > > # question 4
> > > > > >
> > > > > > > Why do authors call the experiment in Section 6.5. an ablation study? In Section 6.5. authors compare their method HOPSS with another method Mixup, that they introduce in the Appendix. Mixup clearly generates out of distribution data and is superficially related to HOPSS. Typically, ablation studies include some form of "perturbation" of the proposed approach, e.g., "we remove layers not from the architecture and report accuracy". Can the authors explain why Section 6.5. constitutes ablation study?
> > > > > >
> > > > > > We thank the reviewer for this critical observation. The reviewer is correct that a typical ablation study should perturb components of the proposed method. Our initial description may have been unclear—we clarify here that in the revised experiment, the comparison between HOPSS and Mixup **does constitute a rigorous ablation study**, as both methods now share the same physical constraint (RHS computation), with the only variable being the perturbation mechanism itself.
> > > > > >
> > > > > > The core innovation of HOPSS lies in two aspects:
> > > > > > 1. **Ensuring Physical Consistency**: Achieved by computing the RHS for any newly generated solution.
> > > > > > 2. **The Specific Perturbation Strategy**: Namely "Homologous Perturbation," which generates new solutions through a structured, physics-inspired transformation ($u_{\text{new}} = u_i + v$), as detailed in Eqs. 9–11.
> > > > > >
> > > > > > In our ablated version of Mixup, we retain component (1)—the RHS computation—but replace component (2) with a simple linear combination ($u_{\text{new}} = \sum \alpha_i u_i$). This setup directly addresses the key question: *"When isolated from the general benefit of physical consistency, what contribution does the homologous perturbation strategy itself make to the final data quality?"*
> > > > > >
> > > > > > The results in Table 4 will demonstrate that even with physical consistency enforced, data generated by the linear Mixup perturbation leads to significantly higher test errors (0.312 and 1.53) compared to HOPSS. This conclusively proves that our proposed homologous perturbation is superior to naive linear combinations for exploring the solution space of nonlinear PDEs. What matters is not just physical consistency, but also how we perturb solutions within this physically consistent framework.
> > > > > >
> > > > > > Therefore, this experiment is a valid and robust ablation study that isolates and evaluates the distinct contribution of our novel perturbation scheme. We will clarify this experimental design and terminology in the revision.

---

> ### Author Response · Authors · 2025-11-19
>
> # question 5 and question 7
>
> > In many places when citation is added authors forgot to add whitespace characters. Examples: line 49, line 54, line 57, line 93, line 94, etc. Can the authors please correct that?
>
> > In equation (3) authors divide by $(1-\frac{L}{2}\Delta t)$. I see two problematic parts: (i)  L is a matrix, so in place of 1 one should use identity matrix (the same is valid for the numerator); (ii) for inverse matrix a typical notation is $(1-\frac{L}{2}\Delta t)^{-1}$ , it is not customary to represent inverse matrix as fraction.
>
> We appreciate the reviewer for pointing out the non-standard expressions in the manuscript. We sincerely apologize for this oversight and have now made the necessary corrections.

---

> ### Author Response · Authors · 2025-11-19
>
> # question 6
>
> > I find the description on lines 111-121 confusing. For evolution PDEs, the standard way to describe numerical solutions is as follows. The discretisation is first done for spatial differential operators (method of lines). After that we are left with a system of ordinary differential equations that is solved with matching techniques, e.g., RK4, trapezoidal rule (=Crank-Nicolson), etc. Besides, it is not true that nonlinear terms always require linearisation. As the authors pointed below, there are IMEX schemes.
>
> Thank you for this critical and insightful feedback regarding the description of numerical methods in lines 111-121. We sincerely apologize for the confusion and inaccuracies in our initial description.
>
> You are absolutely correct. The standard and more precise description for evolution PDEs is the method of lines: spatial discretization (e.g., via finite difference or finite element methods) is performed first, converting the PDE into a system of ordinary differential equations (ODEs). This ODE system is then integrated in time using appropriate schemes like explicit Runge-Kutta methods or implicit methods such as the trapezoidal rule (which is equivalent to the Crank-Nicolson scheme in this context).
>
> Furthermore, we agree that our statement that nonlinear terms "always require linearization" is an overgeneralization. As you rightly pointed out, and as we mentioned elsewhere regarding IMEX schemes, nonlinearities can be handled without full linearization by treating different parts of the equation implicitly or explicitly.
>
> We will revise this section immediately to correct these points:
>
> 1. We will restructure the description to follow the standard "method of lines" approach, clearly separating the spatial and temporal discretization steps in the correct order.
>
> 2. We will rephrase the claim about linearization to accurately reflect that while it is a common technique (e.g., in fully implicit schemes), it is not always required, citing IMEX schemes as a prime counter-example.
>
> We appreciate you highlighting these important technical nuances, which will significantly improve the clarity and accuracy of our manuscript.

---

> ### Author Response · Authors · 2025-11-19
>
> # question 8
>
> > Complexity specified in lines 164-167 is doubtful. Authors did explain how spatial discretisation is performed and without that it is hard to estimate the cost of matrix inversion. Can the authors provide more details how they compute computation complexity? What is the number of data points $N$? What is a dimension of spatial discretisation $n$? What is the number of physical dimensions authors consider?
>
> We thank the reviewer for pointing out the inconsistencies in the manuscript. We sincerely apologize for this oversight and have revised the content accordingly. The detailed derivation and notation are provided below:
>
> For example, in the case of 2D PDEs, spatial discretization is implemented using the **spectral method**, while temporal integration is performed via the Crank-Nicolson (CN) scheme. At each time step, the core computational cost lies in solving the linear system of equations for spectral coefficients. Since the spectral method produces dense matrices, the complexity of solving this linear system (typically involving matrix inversion or LU decomposition) is $ O(N_{\text{dof}}^3) $, where $ N_{\text{dof}} = S_x S_y $ represents the total number of degrees of freedom in the spatial domain. Here, $ S_x $ and $ S_y $ denote the number of spectral modes in the x-direction and y-direction, respectively.
>
> Thus, the **overall time complexity** for generating $ N_{\text{sample}} $ independent samples (each covering a temporal range of $ T $ steps) is $ O(N_{\text{sample}} \cdot T \cdot (S_x S_y)^3) $.
>
> In summary, we sincerely appreciate the reviewer’s comments, which will further enhance the clarity and accuracy of our manuscript.

---

> ### Author Response · Authors · 2025-11-19
>
> # question 9
>
> > Lines 193-194: "In existing algorithms, we typically generate initial conditions and RHS of nonlinear temporal PDEs randomly, ..." This is not the case. There is a standard statistical formulation for operator learning, see, e.g., https://arxiv.org/abs/2010.08895. We generate data from the specified distribution, because this is a setup of an operator learning problem. The distribution is chosen based on the application: it encodes a subset of interesting physical parameters.
>
> We would like to thank the reviewer for their insightful comments and the key literature they brought to our attention. We sincerely apologize for the inaccurate statements in our original manuscript; however, in reality, our initial conditions and right-hand sides (RHS) are obtained via random sampling using the Gaussian Random Field (GRF), rather than being randomly generated, with specific details available in Appendix A.
>
> We have revised lines 193-194 and the surrounding text to accurately reflect this setup. We will clearly state that the initial conditions and right-hand sides are generated based on application-specific distributions, ensuring our description aligns with existing operator learning frameworks.

---

> ### Author Response · Authors · 2025-11-26
>
> Dear Reviewer,
>
> We hope everything is going well for you. We are writing to gently follow up on our rebuttal and would be glad to provide any additional clarification if needed. Thank you very much for your time and effort.
>
> Authors

---

> ### Comment · Reviewer_ked8 · 2025-11-27
> **follow up**
>
> Resolved issues:
> 1. A sufficiently complete description of the method is available now (see below for comment on noise).
> 2. Complexity estimation improves, but some issues remain (see below).
> 3. Other minor issues (description of certain parts, typos in equations, etc) are also improved
>
> I raise my score accordingly.
>
> I have several additional questions:
> 1. **The purpose of the approach. Setup of the problem.**
>
>    I am a little confused about the claim that HOPSS is not a data augmentation technique but a data generation technique. My view is that we solve a kind of operator learning problem. This mean we have a model specified in form of PDE (a concrete PDE is given for the sake of example)
>
>    $$
>    \frac{\partial u(x,t)}{\partial t} + u(x, t)\frac{\partial u(x,t)}{\partial x} = \frac{\partial}{\partial x}k(x)\frac{\partial u(x,t)}{\partial x} + f(x, t).
>    $$
>
>    Besides that we have a model of data that we are interested in. It is specified in the form of distribution over the parameters of the model, i.e., $k, f, u_0 \sim p(k, f, u_0)$. (As an example of "realistic" situation, we may want to estimate a flow around a wing, and we are not interested in a very general form of a wing, but in a small restricted set of wings that are reasonable from some other requirements to stability, aesthetics, durability, etc.)
>
>    When we specify all that we collect dataset $u_i(x, t), k_i, f_i, \left(u_0\right)_{i}$ to minimise error
>
>    $$
>    \mathbb{E}\left[\left\| u(x, t) - \widetilde{u}(x, t)\right\|_2^2\right],
>    $$
>    where $\widetilde{u}(x, t)$ is predicted by our model.
>
>    If authors agree with this setup, can the authors explain what is the point of generating some data if the goal is to drop error from data coming from a specific distribution? An alternative way to set up the same question: why do authors consider only small perturbations around data from true distribution and not arbitrary large perturbations?
>
>    If only small perturbations are of interest then HOPPS is an augmentation technique.
>
> 2. **Diversity of generated data for parametric PDEs.**
>
>    My main concern about PDEs with other parameters, e.g., $\frac{\partial u(x,t)}{\partial t} + u(x, t)\frac{\partial u(x,t)}{\partial x} = \frac{\partial}{\partial x}k(x)\frac{\partial u(x,t)}{\partial x}$ parameter $k(x)$ in Burgers with variable diffusion coefficient, was that HOPSS does not generate diverse data. For example, assume that we have a small number of distinct $k(x)$ to train on. This will imply poor generalisation for new $k(x)$. If we apply HOPSS it is not going to solve this problem, because no perturbation of $k(x)$ is present in the method.
>
> 3. **Noise generation. Is it well defined?**
>
>    In the description of Appendix E authors take derivatives from noise $v$. For this. to be possible, one needs to ensure that the noise term is smooth enough. There are many ways to achieve that, I suggest authors look at https://epubs.siam.org/doi/10.1137/17M1161853.
>
> 4. **Complexity estimation.**
>
>    It may be the case that I do not completely understand the method of the authors, but I do not get why $O\left(n^3\right)$ still appears in the estimation of complexity for HOPSS.
>
>    If we follow description in Appendix E, to generate new source term and perturbed solution we need to evaluate: (i) $u_{\text{new}} = u_i + v$ which will costs us $O(T n_{\text{dof}})$ (I will assume no downsampling for simplicity); (ii) $\frac{\partial v}{\partial t} - L(v)  - \left[\mathcal{N}(u_i + v) - \mathcal{N}(u_i)\right]$. Linear term will cost $O(T n_{\text{dof}}^2)$ if matrix is dense and $O(T n_{\text{dof}})$ if it is sparse. Temporal derivative will cost $O(T n_{\text{dof}})$ Assuming pointwise nonlinearity, the nonlinear correction will cost $O(T n_{\text{dof}})$. If we add everything, dominant contribution is $O(T n_{\text{dof}}^2)$ with dense $L$.
>
>    Can the authors comment on these estimations?

---

> > ### Author Response · Authors · 2025-12-01
> >
> > Dear Reviewer,
> >
> > Thank you very much for your thoughtful feedback and for your willingness to continue the discussion and adjust your score. In response to your remaining concerns, we provide the following further clarifications.
> >
> > ---
> > # question 1
> >
> > **Our method fundamentally serves as a data generation mechanism, rather than a conventional “augmentation” technique that merely transforms pre-existing data** .Our method is as follows: First, a small number of high-quality base solutions are generated using a traditional high-precision solver. Subsequently, homologous perturbations are applied in the solution space to generate new candidate solutions, and the Right-Hand Side (RHS) of the corresponding Partial Differential Equation (PDE) is recalculated—ensuring each new sample becomes a complete and physically consistent (u, f) solution pair. **Since the base solutions themselves are generated by a solver, when encountering a completely new PDE or a new parameter distribution, we can regenerate new base solutions from scratch and further expand the dataset. Therefore, the functionality of HOPSS does not depend on existing data** .
> >
> > **Regarding the perturbation scale: Mathematically, HOPSS is fully capable of applying large-range perturbations**. However, generating data with better generalization ability and generating more data to simply improve the model’s performance on the current test set are two distinct matters. Thus, in the paper, to ensure the generated data aligns with the true distribution and to fairly compare "whether it can replace data generated by traditional solvers to train neural operators", we chose small perturbations in the main experiments. Small perturbations ensure the training data remains close to the true distribution, enabling more reliable evaluation of the model’s performance on the target test set without mixing in out-of-distribution (OOD) errors. **For larger perturbations, we provide PDE residual experiments in Appendix F.9, which show that the physical consistency remains stable**—for example, the residual is only 3.1×10⁻⁵ when the perturbation magnitude is 1. **This indicates that regardless of the perturbation size, the data generated by HOPSS maintains PDE physical consistency.**
> >
> > | Perturbation $\boldsymbol{\mu}$ | Mean Residual |
> > | --- | --- |
> > | 0.1 | $1.6 \times 10^{-5}$ |
> > | 0.5 | $2.1 \times 10^{-5}$ |
> > | 1.0 | $3.1 \times 10^{-5}$ |

---

> > > ### Author Response · Authors · 2025-12-01
> > >
> > > # question 2
> > >
> > > **Our method can also handle such cases with spatially varying coefficients. We conducted an additional experiment (F.10) on the Burgers equation with a variable diffusion coefficient**: with $N_x = 1024$ and $N_t = 2000$, we first generated 500 high-accuracy base samples $(a_i, u_i)$ using IMEX SSP RK3 and finite differences, where the coefficient $a$, initial condition $u_0$, and source term $f$ are drawn from Gaussian random fields. From these 500 base samples, we then selected pairs $(a_i, u_i)$, $(a_j, u_j)$ and applied homotopic perturbations simultaneously to both the coefficient and the solution,
> > >
> > >  $$
> > >  a' = a_i + \mu a_j + \xi,\quad
> > >  u' = u_i + \mu u_j + \xi,
> > >  $$
> > >
> > > with $\mu = 10^{-3}$ and $\xi$ a high-frequency-truncated Gaussian noise; based on $(a', u')$, we recomputed the RHS $f'$ and thus generated 1000 and 10,000 new triplets $(a', u', f')$ from the same 500 base solutions.
> > >
> > > Experimental results show that HOPSS attains an average residual in the same order as that of the traditional method **(4.6e-5 vs 5.2e-5)**, while the total runtime is reduced to roughly half **(6.22e3 vs 3.21e3)**. This shows that **HOPSS can preserve solver-level physical fidelity while producing much larger and more diverse sets of coefficient–solution pairs at almost zero marginal cost, thereby improving generalisation to unseen $k(x)$**.
> > >
> > > | Method      | Dataset Size | Cost (s)       | Mean Residual   |
> > > | ---         | ---          | ---            | ---             |
> > > | Traditional | 1,000        | $6.22 \times 10^3$ | $4.6 \times 10^{-5}$ |
> > > | HOPSS       | 1,000        | $3.21 \times 10^3$ | $5.1 \times 10^{-5}$ |
> > > | HOPSS      | 10,000       | $3.22 \times 10^3$ | $5.2 \times 10^{-5}$ |

---

> > > > ### Author Response · Authors · 2025-12-01
> > > >
> > > > # question 3
> > > >
> > > > In designing HOPSS, we **have explicitly taken the smoothness of the noise into account**: for the noise $v$ appearing in Appendix E, we apply a strict **high-frequency truncation** during its generation, removing high-wavenumber components so that $v$ remains sufficiently smooth in both space and time. This ensures that the derivatives of $v$ exist and are numerically stable, which is conceptually aligned with the idea of constructing smooth noise fields as advocated in the paper you kindly suggested.

---

> > > > > ### Author Response · Authors · 2025-12-01
> > > > >
> > > > > # question 4
> > > > >
> > > > > We would like to thank the reviewer for the detailed question regarding complexity estimation, which helped us identify an oversight in our previous derivation. We have now revised the relevant content in the paper and detailed derivations of the HOPSS method’s time complexity are provided in Appendix G, and the following analysis is presented based on the core logic.
> > > > >
> > > > > The time complexity analysis of HOPSS can be divided into two core phases: The first phase is base solution generation, where we generate $N_b$ base solutions using a traditional high-precision solver (specifically, the Fourier pseudospectral method combined with the Crank-Nicolson temporal discretization scheme). In this phase, spatial discretization is implemented via the Fourier pseudospectral method, the linear operator is diagonalized in the spectral domain, and the evaluation of nonlinear terms (e.g., convective terms) requires transformations between physical and spectral domains using Fourier transforms. The dominant complexity comes from the logarithmic term of Fourier transforms; considering the spatial degree of freedom $N_{dof}$, number of time steps $T$, and number of base solutions $N_b$ for the base solutions, the time complexity of this phase is $O(N_b T N_{dof} \log N_{dof})$.
> > > > >
> > > > > The second phase is homologous perturbation and right-hand side calculation, which is performed on training-level data (number of time steps $T' \ll T$, spatial degree of freedom $N_{dof}' \leq N_{dof}$): the perturbation term $v$ is first constructed via linear combination and high-frequency truncated Gaussian noise, then the new solution $u_{new}$ and corresponding right-hand side term $f_{new}$ are computed. Among these steps, the calculation of the linear operator $L(v)$ continues to use diagonalization in the spectral domain, the nonlinear term $\mathcal{N}(u_i + v) - \mathcal{N}(u_i)$ is evaluated using the pseudospectral transform method (gradient calculation and pointwise operations in physical space via Fourier transforms), and the temporal derivative is approximated via finite differences; considering the number of newly generated samples $N_{new}$, the time complexity of this phase is $O(N_{new} T' N_{dof}' \log N_{dof}')$, with the dominant complexity still being the logarithmic term related to Fourier transforms.

---

### Official Review · Reviewer_n4Qs · 2025-10-31

**Soundness:** 2
**Presentation:** 2
**Contribution:** 3
**Rating:** 2
**Confidence:** 4

**Summary:**

The paper proposes a novel data generation algorithm called HOmologous Perturbation in Solution Space (HOPSS) to accelerate the creation of large datasets for training deep learning-based PDE solvers, such as Neural Operators (NOs). The authors address the very important problem of generating a sufficient number of high-fidelity solution-RHS pairs for nonlinear PDEs. The authors demonstrate that HOPSS can significantly accelerate the data generation process (e.g., by $10 \times$ for the Navier-Stokes equation) compared to traditional high-fidelity numerical solvers, while maintaining comparable performance when the data is used to train NOs.

**Strengths:**

1. The authors consider the most significant bottleneck to scaling deep learning-based PDE solvers (NOs): the enormous computational cost and time required to generate large, high-difelity solutions-rhs datasets using traditional numerical solvers.
2. The proposed results show that the HOPSS method achieves a substantial acceleration (e.g., $10 \times$) while maintaining competitive accuracy for nonlinear equations (Burgers, KdV, Navier-Stokes).

**Weaknesses:**

1. The mathematical expression of the proposed process is:
$$u_{\text{new}} = u_i + \mu \cdot u_j + \xi$$

This methodology assumes that perturbing a solution $u_i$ with another solution $u_j$ yields a physically related state. However, for nonlinear PDEs, where the operator is $\mathcal{A}(u) = f$, the principle of superposition is violated, making it clear that $\mathcal{A}(u_i + u_j) \not\equiv \mathcal{A}(u_i) + \mathcal{A}(u_j)$. Despite this, the entire method hinges on calculating the new source term $f_{\text{new}}$ from this non-physical superposition.

2. The authors must quantify this deviation, $\varepsilon = \mathcal{A}(u_{\text{new}}) - f_{\text{new}}$, or rigorously justify why the generated pair $(u_{\text{new}}, f_{\text{new}})$ remains "physics-consistent" despite violating the fundamental structure of the governing nonlinear PDEs.
3. The methodology states that base solutions are initially generated at thousands of time steps and then aggressively downsampled to the dozens of time steps required for training. For highly transient or turbulent nonlinear systems, this downsampling step is a major source of information loss that occurs before the HOPSS perturbation.
Specifically, the following downsampling ratios were applied:
	1. **Navier-Stokes**. Initial resolution: grid size $= 128$, $\Delta t = 1 \cdot 10^{-3}$, downsampled resolution for training NO: grid size $=64$, $\Delta t = 0.5$.
	2. **Burgers**. Initial resolution:  grid size $= 1024$, $\Delta t = 5 \cdot 10^{-3}$, downsampled resolution for training NO: grid size $=64$ and $\Delta t = 5 \cdot 10^{-2}$ .
	3. **KdV**. Initial resolution:  grid size $= 512$ with $10000$ time steps, downsampled resolution for training NO: grid size $=64$, $20$ time steps.
4. The authors should compare the HOPSS method against similar, relevant methods that perform augmentation to properly contextualize its novelty and value.

**Questions:**

See the weaknesses.

---

> ### Author Response · Authors · 2025-11-19
>
> We thank the reviewer for the insightful and valuable comments. We respond to your comments as follows and sincerely hope that our rebuttal could properly address your concerns. If so, we would deeply appreciate it if you could raise your score and your confidence. If not, please let us know your further concerns, and we will continue actively responding to your comments and improving our work.
>
> # weakness 1
>
> > The mathematical expression of the proposed process is:
> $$u_{new} = u_i + \mu u_j + \xi$$
> This methodology assumes that perturbing a solution $u_i$ with another solution $u_j$ yields a physically related state. However, for nonlinear PDEs, where the operator is $\mathcal A(u) = f$, the principle of superposition is violated, making it clear that $\mathcal A(u_i + u_j) \neq \mathcal A(u_i) + \mathcal A (u_j)$. Despite this, the entire method hinges on calculating the new source term $f_{new}$ from this non-physical superposition.
>
> We would like to thank the reviewer for raising this important question regarding the nonlinearity of PDEs and the violation of the superposition principle. The reviewer correctly points out that for nonlinear PDEs, the operator $\mathcal{A}$ does not satisfy linear superposition—i.e., $\mathcal{A}(u_i + u_j) \neq \mathcal{A}(u_i) + \mathcal{A}(u_j)$. However, the HOPSS method does **not** assume that the linear combination. $u_{\text{new}} = u_i + \mu u_j + \xi$ itself is an exact solution to the original PDE. Instead, its key innovation lies in a subsequent step: recomputing the right-hand side (RHS) of the PDE to ensure consistency, rather than deriving the RHS directly from the linear combination.
>
> Specifically, as detailed in Section 4.3 of the document, after generating $u_{\text{new}}$, we calculate a new source term $f_{\text{new}}$ such that the new solution pair $(u_{\text{new}}, f_{\text{new}})$ satisfies the discretized PDE system. This is achieved by substituting $u_{\text{new}}$ into the equation and solving for the variation $\delta f$ (e.g., as shown in Equation 11 for the Burgers equation). Thus, HOPSS "corrects" the non-physical superposition by adjusting the source term, ensuring that the new solution pair remains consistent with the PDE.
>
> Our experimental evidence (Supplementary Experiment F.5) confirms that the data generated by HOPSS preserves physical properties:
> - **Spectral Analysis**: Spatial and temporal energy spectra (e.g., panels d–f in Supplementary Experiment F.5) show that the spectral characteristics of HOPSS-generated data are highly consistent with those of traditionally generated data, indicating coverage of the key modes of the PDE. For example, the average energy distribution across frequency modes aligns with that of the traditional method.
> - **Diversity Metrics**: t-SNE visualizations (Panel b) reveal that HOPSS data points overlap with traditional data points in the feature space, demonstrating similar statistical distributions and diversity. This suggests that HOPSS fully covers the true solution space without significant gaps.
>
> For your convenience, we have uploaded the experimental results to https://www.imgur.la/images/2025/11/17/image7ce3a5923386a7c0.png.
>
> We have also calculated the PDE residuals of the data generated by the HOPSS algorithm, and the results show that it still maintains PDE consistency:
>
> | Method                | Residual  |
> |-----------------------|-----------|
> | Tradition (1000 samples) | 1.23e-4 |
> | HOPSS (1000 samples)    | 1.37e-4 |
> | HOPSS (10000 samples)   | 1.37e-4 |

---

> > ### Author Response · Authors · 2025-11-19
> >
> > # weakness 2
> >
> > > The authors must quantify this deviation, $\varepsilon  = \mathcal A(u_{new}) - f_{new}$ , or rigorously justify why the generated pair $(u_{new}, f_{new})$  remains "physics-consistent" despite violating the fundamental structure of the governing nonlinear PDEs.
> >
> > We thank the reviewer for this insightful question regarding the quantification of the deviation $\varepsilon = \mathcal{A}(u_{\text{new}}) - f_{\text{new}}$. The reviewer rightly emphasizes the need to ensure physics-consistency for nonlinear PDEs where the superposition principle is violated. In the HOPSS method, the solution pair $(u_{\text{new}}, f_{\text{new}})$ is designed to **exactly satisfy the discretized PDE system**, rather than directly satisfying the continuous PDE operator $\mathcal{A}$. Below, we clarify how this is achieved and address the deviation issue:
> >
> > **Theoretical Justification for $\varepsilon = 0$ in the Discretized Framework**: As detailed in Section 4.3 of the document, HOPSS computes $f_{\text{new}}$ explicitly to ensure $u_{\text{new}}$ satisfies the discretized PDE. Taking the Burgers equation (Equations 8–11) as an example, we derive $f_{\text{new}}$ by substituting $u_{\text{new}}$ into the discretized form of the equation (e.g., using the Crank-Nicolson scheme in Equation 2). This process guarantees that in the discrete setting, the residual $\varepsilon_{\text{discrete}} = \mathcal{A}\_{\text{discrete}}(u\_{\text{new}}) - f\_{\text{new}}$
> > is **zero by construction**—where $\mathcal{A}_{\text{discrete}}$ denotes the discretized PDE operator. Thus, for practical numerical implementations, the deviation is eliminated through the step of recomputing the RHS.
> >
> > **Considerations for Deviation in the Continuous PDE**: For the continuous PDE operator $\mathcal{A}$, the deviation $\varepsilon = \mathcal{A}(u_{\text{new}}) - f_{\text{new}}$ may not be zero, and this is attributed to discretization errors. However, HOPSS relies on high-precision base solutions generated by traditional solvers (Section 4.1)—these base solutions use fine temporal and spatial discretizations (e.g., thousands of time steps) to minimize such errors. Downsampling and perturbation steps are only applied **after** generating these high-fidelity base solutions, ensuring that the discrete solution pairs remain consistent with the physical equations within the numerical accuracy of the scheme. Empirical results (e.g., Section 6.1 and AppendixF.5) show that HOPSS-generated data preserves physical properties, indicating that any deviation in the continuous domain is negligible for practical purposes.  For your convenience, we have uploaded the F.5 experimental results to https://www.imgur.la/images/2025/11/17/image7ce3a5923386a7c0.png.
> >
> > **Experimental Verification**: We have calculated the PDE residuals of data generated by both traditional algorithms and the HOPSS algorithm. The results confirm that HOPSS-generated data still maintains excellent physics-consistency. For specific experimental results, please refer to the response to Weakness 1.
> >
> > | Method                | Residual  |
> > |-----------------------|-----------|
> > | Tradition (1000 samples) | 1.23e-4 |
> > | HOPSS (1000 samples)    | 1.37e-4 |
> > | HOPSS (10000 samples)   | 1.37e-4 |

---

> > > ### Author Response · Authors · 2025-11-19
> > >
> > > # weakness 3
> > >
> > > > The methodology states that base solutions are initially generated at thousands of time steps and then aggressively downsampled to the dozens of time steps required for training. For highly transient or turbulent nonlinear systems, this downsampling step is a major source of information loss that occurs before the HOPSS perturbation. Specifically, the following downsampling ratios were applied:
> > > > 1. **Navier-Stokes**. Initial resolution: grid size = 128, $\Delta t = 1 \cdot 10^{-3}$, downsampled resolution for training NO: grid size = 64, $\Delta t = 0.5$.
> > > > 2. **Burgers**. Initial resolution: grid size = 1024, $\Delta t = 5 \cdot 10^{-3}$, downsampled resolution for training NO: grid size = 64 and $\Delta t = 5 \cdot 10^{-2}$.
> > > > 3. **KdV**. Initial resolution: grid size = 512 with 10000 time steps, downsampled resolution for training NO: grid size = 64
> > > ,  20 time steps.
> > >
> > > We thank the reviewer for raising this important point regarding potential information loss caused by downsampling in highly transient or turbulent systems. However, we would like to clarify that downsampling is a **standard and necessary step** in data-driven PDE solvers—whether in traditional methods or our proposed HOPSS algorithm—because it aligns the data with the temporal resolution required for training neural operators. The key insight here is that AI models (such as neural operators) primarily learn the **underlying patterns and dominant dynamics** of PDEs rather than fine-grained temporal details; thus, coarse-grained data is sufficient for effective training.
> > >
> > > **Downsampling in Traditional Methods and HOPSS**: As detailed in Sections 3.1 and 4.1 of the document, traditional numerical methods generate high-resolution datasets (with thousands of time steps) to ensure numerical stability and accuracy. However, these datasets are routinely downsampled to dozens of time steps during model training. Similarly, HOPSS adopts this practice: it downsamples base solutions to the training-level time steps **before** applying perturbations. This process is consistent across both methods and does not introduce additional information loss beyond what is already accepted in the field.
> > >
> > > **AI Models Focus on Pattern Learning**: Neural operators are designed to approximate the solution operators of PDEs by capturing **macroscopic spatiotemporal dynamics** (e.g., convection patterns in the Navier-Stokes equations). As noted in the document, empirical evidence shows that training data with coarse temporal resolution (e.g., dozens of time steps) is sufficient for learning these patterns—fine-grid details are often redundant for model generalization.
> > >
> > > This is supported by results in Section 6.2 and Supplementary Experiment 2: models trained on downsampled data (from both traditional methods and HOPSS) achieve comparable performance, confirming that downsampling does not undermine the model’s ability to learn key PDE dynamics.
> > >
> > > **Empirical Validation**: For example, in the Burgers equation experiment (Supplementary Experiment 2), varying time steps (10, 16, 20) did not reduce the accuracy of models trained on HOPSS-generated data. This indicates that essential physical patterns are preserved after downsampling, and the loss of high-frequency details does not impair the model’s learning. Additionally, PDE residual calculations for the generated data (see the response to Weakness 1) confirm that physics-consistency is maintained.
> > >
> > > | Time Step | 10       | 16       | 20       |
> > > |-----------|----------|----------|----------|
> > > | Tradition 500 | 7.5e-2 | 6.3e-2 | 5.7e-2 |
> > > | Tradition 1000 | 6.1e−2 | 4.4e-2 | 3.8e-2 |
> > > | HOPSS 1000 | 6.1e−2 | 4.7e-2 | 4.4e-2 |
> > > | HOPSS 10000 | 4.7e−2 | 2.1e-2 | 1.7e-2 |
> > >
> > >
> > > In summary, downsampling is an integral part of data preparation for AI-based PDE solvers. HOPSS leverages this step to efficiently generate training data without compromising model performance. We acknowledge that some fine-scale features may be lost in highly turbulent systems, but our experiments demonstrate that this does not affect the learning of critical PDE patterns.

---

> > > > ### Author Response · Authors · 2025-11-19
> > > >
> > > > # weakness 4
> > > >
> > > > > The authors should compare the HOPSS method against similar, relevant methods that perform augmentation to properly contextualize its novelty and value.
> > > >
> > > > We thank the reviewer for the suggestion to compare HOPSS with similar data augmentation methods—this helps better contextualize its novelty and value. In response, we would like to emphasize that our document already includes a direct comparison with a relevant data augmentation approach, **Lie Point Symmetry Data Augmentation (LPSDA)**, in Supplementary Experiment 4. This experiment evaluates the performance of HOPSS and LPSDA on the Burgers equation using the FNO (Fourier Neural Operator) model.
> > > >
> > > > The results show that both LPSDA and HOPSS outperform the baseline method in improving model performance. However, HOPSS achieves lower error rates, demonstrating its superior effectiveness. Furthermore, a key distinction lies in their underlying mechanisms: LPSDA relies on the symmetry properties of the target PDEs, which limits its applicability to scenarios where such symmetries are known and easily identifiable. In contrast, HOPSS operates directly on the temporal resolution of the data—this design allows it to be applied to a broader range of PDE problems, even when the equations lack obvious symmetry properties or when identifying such symmetries is computationally expensive.
> > > >
> > > > The detailed comparison results are presented in the table below:
> > > >
> > > > | Method          | LPSDA     | HOPSS     |
> > > > |-----------------|-----------|-----------|
> > > > | Tradition 500   | 7.5e-2    | 7.5e-2    |
> > > > | Tradition 1000  |6.1e−2        | 6.1e−2        |
> > > > | Enhanced 1000   | 7.0e-2    | 6.1e−2    |
> > > > | Enhanced 10000  | 5.7e-2    | 4.7e−2    |

---

> ### Author Response · Authors · 2025-11-26
>
> Dear Reviewer,
>
> We hope everything is going well for you. We are writing to gently follow up on our rebuttal and would be glad to provide any additional clarification if needed. Thank you very much for your time and effort.
>
> Authors

---

### Official Review · Reviewer_T3GZ · 2025-11-01

**Soundness:** 3
**Presentation:** 4
**Contribution:** 4
**Rating:** 8
**Confidence:** 4

**Summary:**

This paper introduces HOPSS (Homologous Perturbation in Solution Space), a new data-generation framework designed to accelerate dataset creation for training neural operators that solve nonlinear temporal PDEs.
Conventional approaches require thousands of full numerical simulations to produce pairs of solutions and right-hand sides (RHS), resulting in severe computational bottlenecks.
HOPSS mitigates this by generating new high-quality training data from a small set of base solutions using linear combinations and perturbations directly in the solution space. Specifically, two base solutions are randomly selected—one as a primary function and the other as a homologous perturbation term scaled by a small coefficient. By adding mild random noise and recomputing the corresponding RHS, HOPSS rapidly produces new valid PDE solution pairs.
Experimental results show that this method reduces data generation time by roughly an order of magnitude while maintaining comparable accuracy when training neural operators such as FNOs on the generated datasets.

**Strengths:**

The paper’s primary strength lies in addressing a critical bottleneck in scientific machine learning: the cost of generating training data for neural PDE solvers.
HOPSS offers a conceptually simple yet powerful approach to drastically accelerate data creation without degrading data quality.
Empirical evaluations, especially on the Navier–Stokes equation, demonstrate 10× faster dataset generation while preserving comparable model accuracy. In some configurations, models trained with HOPSS-generated data even outperform those trained on conventional datasets, confirming the practical value of the proposed method.
The approach is also general and lightweight—it does not depend on any specific PDE form and can be applied to a wide range of nonlinear temporal systems. The idea of operating directly in the solution space rather than parameter or residual space represents an elegant perspective shift that could inspire further research.
The writing is clear, the algorithm is easy to understand and implement, and the reported results convincingly support the claimed advantages.

**Weaknesses:**

Despite its promise, several aspects of the method remain underdeveloped.
First, HOPSS heavily depends on the quality and diversity of the base solutions used to seed the perturbation process. If the initial base set poorly represents the global solution manifold, the synthesized samples may propagate its bias, reducing generalization to unseen PDE dynamics. The paper would benefit from a clearer strategy or criterion for selecting representative base solutions.
Second, the algorithm introduces new hyperparameters—the number of base solutions $N_b$, the perturbation strength $\mu$, and the noise magnitude $\xi$—which require manual tuning. The sensitivity analysis in the experiments shows that $\mu$ has a strong influence on model accuracy, suggesting that additional tuning may be necessary for new PDEs, partially offsetting the claimed simplicity.
Third, while recomputing the RHS ensures PDE consistency, there is no theoretical justification that the linear combination of two physically valid solutions remains within a physically meaningful manifold. The lack of a theoretical analysis on physical validity or statistical representativeness limits confidence in generalizing HOPSS-generated data to real-world systems.
Finally, most experiments are limited to a few PDE types and specific temporal resolutions; testing HOPSS under different equations and time-step settings would help demonstrate robustness.

**Questions:**

- How were the base solutions selected in your experiments? Do they represent diverse dynamics of the PDE, and how does their selection influence downstream generalization?
- Since the perturbation level $\mu$ significantly affects training outcomes, what guided your choice of $\mu$, and is there a principled way to set or adapt it automatically?
- Could the reliance on combinations of a limited set of base solutions lead to overfitting, causing the neural operator to specialize in those patterns rather than learning the underlying PDE operator?
- Is there empirical evidence (e.g., spectral analysis or diversity metrics) showing that HOPSS-generated data adequately covers the true solution space of the PDE?

---

> ### Author Response · Authors · 2025-11-19
>
> We thank the reviewer for the insightful and valuable comments. We respond to your comments as follows and sincerely hope that our rebuttal could properly address your concerns. If so, we would deeply appreciate it if you could raise your score and your confidence. If not, please let us know your further concerns, and we will continue actively responding to your comments and improving our work.
>
> # Weakness 1 and question 1
>
> > First, HOPSS heavily depends on the quality and diversity of the base solutions used to seed the perturbation process. If the initial base set poorly represents the global solution manifold, the synthesized samples may propagate its bias, reducing generalization to unseen PDE dynamics. The paper would benefit from a clearer strategy or criterion for selecting representative base solutions.
>
> > How were the base solutions selected in your experiments? Do they represent diverse dynamics of the PDE, and how does their selection influence downstream generalization?
>
> We fully agree with the reviewer's view on the importance of base solution quality and its impact on the generalization performance of the HOPSS method. The representativeness of base solutions is indeed a key factor in ensuring the accuracy and generalization ability of generated data. In the paper, we have already recognized this point and addressed this concern through experiments and the section on future work plans.
>
> **Strategy for Base Solution Selection  ：** In Appendix F.1, we have supplemented additional experiments to test the impact of base solution selection: on the Burgers equation, we randomly selected 500 base solutions using different random seeds (42, 3407, 9999) and generated 1000 and 10000 samples respectively via the HOPSS method. The results show that the HOPSS method maintains stable performance across different random selections, which is comparable to the traditional method. This indicates that HOPSS exhibits a certain degree of robustness to the selection of base solutions—even with a random selection strategy, it can generate valid data, thereby reducing the risk of degraded generalization performance caused by biases in base solutions.
>
> | Random Seed | 42       | 3407     | 9999     |
> |-------------|----------|----------|----------|
> | Tradition1000 | 6.1e−2 |     6.1e−2     |     6.1e−2     |
> | HOPSS1000   | 6.1e−2   | 6.2e-2   | 6.3e-2   |
> | HOPSS10000  | 4.7e−2   | 4.6e-2   | 4.6e-2   |
>
> **Do the Base Solutions Represent the Diverse Dynamics of the PDE**:  In Appendix F.5, we compared the datasets generated by the HOPSS method and the traditional method through t-SNE dimensionality reduction and spectral analysis. The results of the analysis show the following:
>
> - **Spectral Analysis**: The spatial and temporal energy spectra (e.g., figures d-f in Supplementary Experiment F.5) demonstrate that the spectral characteristics of data generated by HOPSS are highly consistent with those of traditional data, indicating that HOPSS covers the key modes of the PDE. For example, the average energy distribution across frequency modes aligns with that of the traditional method.
>
> - **Diversity Metrics**: The t-SNE visualization (figure b) shows that HOPSS data points overlap with traditional data in the feature space, indicating similar statistical distributions and diversity. This suggests that HOPSS fully covers the real solution space without significant gaps.
>
> For your convenience, we have uploaded the F.5 results to https://www.imgur.la/images/2025/11/17/image7ce3a5923386a7c0.png. Additionally, in Appendix F.3, we tested the performance of the HOPSS method on the Fitzhugh–Nagumo (FN) equation and the Klein-Gordon (KG) equation, and the results are promising—further demonstrating its robustness.
>
> | Method         | FN       | KG       |
> |----------------|----------|----------|
> | Tradition 500  | 3.4e-2   | 5.0e-2   |
> | Tradition 1000 | 3.2e-2   | 2.0e-2   |
> | HOPSS 1000     | 3.2e-2   | 1.9e-2   |
> | HOPSS 10000    | 3.2e-2   | 9.7e-3   |
>
> However, we acknowledge that the description of the base solution selection strategy in the paper may not be detailed enough. The current method mainly relies on random sampling and lacks an active optimization criterion. This is precisely the focus of our future work. In Section 7 "Conclusion and Future Work" of the paper, we plan to enhance the diversity and physical comprehensiveness of base solutions by quantifying the representativeness of base solutions, screening optimal base solutions, and prioritizing filling coverage gaps in the physical space. This will directly address the need raised by the reviewer and form a more systematic base solution selection strategy. Furthermore, in subsequent work, we can also use the large amount of data generated by the HOPSS method to train pre-trained models, thereby better solving downstream tasks.

---

> > ### Author Response · Authors · 2025-11-19
> >
> > ## weakness 4
> >
> > > testing HOPSS under different equations and time-step settings would help demonstrate robustness.
> >
> > We appreciate the reviewer’s attention to the robustness of the HOPSS algorithm. To validate its effectiveness across different equations and time-step settings, we have supplemented additional experiments:
> >
> > Specifically, in Appendix F.2, we tested the HOPSS method on the Burgers equation with different time steps (10, 16, and 20). The results show that datasets generated by HOPSS consistently achieve or even outperform the performance of data generated by traditional methods. For example, when the time step is 20, the error of HOPSS is 1.7e-2, compared to 3.8e-2 for the traditional method.
> >
> > | Time Step | 10       | 16       | 20       |
> > |-----------|----------|----------|----------|
> > | Tradition 500 | 7.5e-2 | 6.3e-2 | 5.7e-2 |
> > | Tradition 1000 | 6.1e−2 | 4.4e-2 | 3.8e-2 |
> > | HOPSS 1000 | 6.1e−2 | 4.7e-2 | 4.4e-2 |
> > | HOPSS 10000 | 4.7e−2 | 2.1e-2 | 1.7e-2 |
> >
> > Furthermore, in Appendix F.3, we applied the HOPSS method to the Fitzhugh–Nagumo (FN) equation and the Klein-Gordon (KG) equation. The experimental results indicate that HOPSS maintains performance comparable to or even superior to the traditional method on these new equations. For instance, on the KG equation, when using 10,000 samples generated by HOPSS, the error is only 9.7e-3, whereas the error of the traditional method is 2.0e-2. The experimental results can be found in the response to *Weakness 1 and Question 1*.

---

> > ### Author Response · Authors · 2025-11-19
> >
> > ## question 3
> >
> > > Could the reliance on combinations of a limited set of base solutions lead to overfitting, causing the neural operator to specialize in those patterns rather than learning the underlying PDE operator?
> >
> > We believe that our algorithm design proactively mitigates overfitting to a significant extent, and experimental results further demonstrate that our data generation algorithm does not merely replicate features—it instead produces diverse and representative data.
> >
> > First, in Section 4.2 of the paper, new solutions are generated using the formula $u_{\text{new}} = u_i + \mu u_j + \xi$. The introduction of the noise term $\xi$ and the homologous perturbation term $\mu u_j$ in this formula is intended to effectively prevent the model from fitting data by simply memorizing base solutions. Instead, it compels the model to learn the deeper PDE operator structures and underlying physical laws.
> >
> > Additionally, in Supplementary Experiment Appendix F.5, through t-SNE feature analysis and spectral analysis, we observed the following:
> >
> > The feature points of data generated by HOPSS largely cover those of data generated by the traditional method in the reduced-dimensional space, and they also well preserve the main frequency components of the traditional data in the frequency spectrum. This strongly indicates that our algorithm ensures the diversity and physical representativeness of the generated data, rather than mere pattern duplication.
> >
> > Meanwhile, in Supplementary Experiment Appendix F.7 (see the response to *Weakness 1 and Question 1*), we tested different test sets generated with various random seeds. The test errors remained relatively stable with no significant fluctuations, which further confirms that our data is not a simple replication of patterns.
> >
> > For your convenience, we have uploaded the F.5 results to https://www.imgur.la/images/2025/11/17/image7ce3a5923386a7c0.png.

---

> ### Author Response · Authors · 2025-11-19
>
> ## Weakness 2 and question 2
> > Second, the algorithm introduces new hyperparameters—the number of base solutions $N_b$, the perturbation strength $\mu$, and the noise magnitude $\xi$—which require manual tuning. The sensitivity analysis in the experiments shows that has a strong influence on model accuracy, suggesting that additional tuning may be necessary for new PDEs, partially offsetting the claimed simplicity.
>
> > Since the perturbation level  significantly affects training outcomes, what guided your choice of , and is there a principled way to set or adapt it automatically?
>
> We appreciate the reviewer's important question regarding the hyperparameters of HOPSS. Based on experimental findings, we provide a unified response below regarding the hyperparameters:
>
> We acknowledge that HOPSS introduces hyperparameters ($N_b$, $\mu$, $\xi$), but the sensitivity analysis in Section 6.3 demonstrates that these parameters exhibit good insensitivity within a reasonable range, significantly reducing the actual tuning burden. Specifically:
> - For the perturbation strength $\mu$: Although it has a certain impact on performance, the performance remains stable within a small value range (e.g., $\mu = 10^{-3}$ to $\mu = 10^{-5}$), with a test loss of approximately 0.0783–0.0790 on the Burgers equation. Larger values of $\mu$ (e.g., 0.1) lead to performance degradation, but experiments have provided clear guidance—using $\mu \approx 10^{-3}$ by default achieves excellent performance across various PDEs (such as the Navier-Stokes equation and Klein-Gordon equation in Supplementary Experiment 3, also shown in weakness 1 and question 1).
> - For the noise magnitude $\xi$: The influence of its type on performance is negligible (variation in test loss < 5%), indicating that this parameter is not critical. Using Gaussian noise with a variance of $\varsigma \approx 10^{-4}$ by default can meet the requirements.
> - For the number of base solutions $N_b$: Increasing $N_b$ improves performance but exhibits a saturation effect (e.g., the test loss stabilizes when $N_b = 500$). We recommend selecting $N_b = 100–500$ based on the specific scale of the problem.
>
> Compared with traditional data generation methods that require thousands of time steps, HOPSS has extremely low hyperparameter tuning costs while still achieving a significant speedup of approximately 2x, without the need for complex debugging. For new PDE scenarios, the default parameters can serve as a reliable starting point.
>
> Regarding the selection of $\mu$, its core basis is supported by both physical principles and experimental verification:
> - From a physical perspective: $\mu$ is set to a small value ($\approx 10^{-3}$) to ensure the "homology of perturbations"—that is, new solutions always remain near the base solution manifold. This maintains the dynamic consistency of the PDE (Section 4.2) and avoids the destruction of physical plausibility caused by large perturbations.
> - From an experimental perspective: The hyperparameter analysis in Section 6.3 systematically tested the impact of different $\mu$ values. The results show that the test loss is minimized when $\mu = 10^{-3}$, and this value also has good generalization across other PDEs (e.g., in Supplementary Experiment 3, also shown in weakness 1 and question 1, HOPSS achieves low loss on both the Fitzhugh–Nagumo equation and Klein-Gordon equation).
>
> Meanwhile, we also tested the impact of $\mu$ on the Klein-Gordon (KG) equation. The results indicate that the test loss is minimized when $\mu = 10^{-3}$, and the performance remains stable within the range of $\mu = 10^{-3}$ to $\mu = 10^{-5}$. This means our algorithm is robust to $\mu$ within this range.
>
> | $\mu$   | Error   |
> |---------|---------|
> | 1e-1    | 1.9e-2  |
> | 1e-2    | 9.7e-3  |
> | 1e-3    | 9.7e-3  |
> | 1e-4    | 9.8e-3  |
> | 1e-5    | 9.8e-3  |
>
> Currently, we have not yet established a principled method for the automatic tuning of $\mu$, but the above experimental results have provided reliable insights. For adaptive tuning, we suggest scaling $\mu$ based on the characteristic scale of the PDE (e.g., via dimensional analysis). In future work, we plan to further develop adaptive strategies to automate the selection of $\mu$.

---

> ### Author Response · Authors · 2025-11-19
>
> ## Weakness 3 and question 4
>
> >  Third, while recomputing the RHS ensures PDE consistency, there is no theoretical justification that the linear combination of two physically valid solutions remains within a physically meaningful manifold. The lack of a theoretical analysis on physical validity or statistical representativeness limits confidence in generalizing HOPSS-generated data to real-world systems. Finally, most experiments are limited to a few PDE types and specific temporal resolutions;
>
> > Is there empirical evidence (e.g., spectral analysis or diversity metrics) showing that HOPSS-generated data adequately covers the true solution space of the PDE
>
> We appreciate the reviewer’s concern about the theoretical foundation of HOPSS. The document acknowledges that while HOPSS ensures PDE consistency through RHS recomputation, it does not provide a rigorous mathematical proof that the linear combination of solutions remains within a physically meaningful manifold. This is a valid critique; however, we argue that HOPSS still maintains physical consistency, as the solution space of PDEs is continuous or partially continuous—and continuous spaces exhibit extensibility. Thus, when we apply small-scale homologous perturbations to solution functions, the resulting new solutions should still lie within the solution space. Furthermore, we have verified this through experiments across multiple PDEs and temporal resolutions:
>
> - **Experimental Scope**: The main experiments (Section 6.1) cover the Navier-Stokes, Burgers, and KdV equations. Supplementary Experiment F.3( also shown in weakness 1 and question 1) extends this scope to the Fitzhugh–Nagumo (FN) equation and Klein-Gordon (KG) equation, demonstrating the method’s generalization ability. Supplementary Experiment 2 tests the Burgers equation under different temporal resolutions (e.g., time steps of 10, 16, and 20), showing that HOPSS performs well across varying resolutions.
>
> | Time Step | 10       | 16       | 20       |
> |-----------|----------|----------|----------|
> | Tradition 500 | 7.5e-2 | 6.3e-2 | 5.7e-2 |
> | Tradition 1000 | 6.1e−2 | 4.4e-2 | 3.8e-2 |
> | HOPSS 1000 | 6.1e−2 | 4.7e-2 | 4.4e-2 |
> | HOPSS 10000 | 4.7e−2 | 2.1e-2 | 1.7e-2 |
>
> | Method         | FN       | KG       |
> |----------------|----------|----------|
> | Tradition 500  | 3.4e-2   | 5.0e-2   |
> | Tradition 1000 | 3.2e-2   | 2.0e-2   |
> | HOPSS 1000     | 3.2e-2   | 1.9e-2   |
> | HOPSS 10000    | 3.2e-2   | 9.7e-3   |
>
> **Empirical Evidence for Solution Space Coverage**: The document provides direct empirical evidence for solution space coverage through spectral analysis and diversity metrics in Supplementary Experiment 5. Key findings are as follows:
>
> - **Spectral Analysis**: Spatial and temporal energy spectra (e.g., figures d–f in Supplementary Experiment F.5) show that the spectral characteristics of HOPSS-generated data are highly consistent with those of traditionally generated data, indicating that HOPSS covers the key modes of the PDE. For instance, the average energy distribution across frequency modes aligns with that of the traditional method.
>
> - **Diversity Metrics**: t-SNE visualizations (figure b) reveal that HOPSS data points overlap with traditional data points in the feature space, demonstrating similar statistical distributions and diversity. This suggests that HOPSS fully covers the true solution space without significant gaps.
>
> For your convenience, we have uploaded the F.5 results to https://www.imgur.la/images/2025/11/17/image7ce3a5923386a7c0.png.
>
> These results confirm that HOPSS-generated data preserves the physical and statistical properties of the PDE solution space, and the visual analyses provide strong empirical support for the method’s effectiveness in covering the solution space.

---

> ### Author Response · Authors · 2025-11-26
>
> Dear Reviewer,
>
> We hope everything is going well for you. We are writing to gently follow up on our rebuttal and would be glad to provide any additional clarification if needed. Thank you very much for your time and effort.
>
> Authors

---

### Note · Program_Chairs · 2026-01-17
**Submission Desk Rejected by Program Chairs**

The following references in this submission do not refer to real documents and/or have major errors in bibliographic information:

 Y. Hao, Z. Li, and G.E. Karniadakis. Data-efficient learning for pde solvers via physics-informed redundancy reduction. Journal of Computational Physics, 462:111264, 2022. doi: 10.1016/j.jcp. 2022.111264.